# *DetectRL*: Benchmarking LLM-Generated Text Detection in Real-World Scenarios

**Junchao Wu**[1]   **Runzhe Zhan**[1]   **Derek F. Wong**[1]*   **Shu Yang**[1]
**Xinyi Yang**[1]   **Yulin Yuan**[2]   **Lidia S. Chao**[1]

[1]NLP[2]CT Lab, Department of Computer and Information Science, University of Macau
[2]Department of Chinese Language and Literature, University of Macau
nlp2ct.{junchao,runzhe,shuyang,xinyi}@gmail.com
{derekfw,yulinyuan,lidiasc}@um.edu.mo

## Abstract

Detecting text generated by large language models (LLMs) is of great recent interest. With zero-shot methods like DetectGPT, detection capabilities have reached impressive levels. However, the reliability of existing detectors in real-world applications remains underexplored. In this study, we present a new benchmark, *DetectRL*, highlighting that even state-of-the-art (SOTA) detection techniques still underperformed in this task. We collected human-written datasets from domains where LLMs are particularly prone to misuse. Using popular LLMs, we generated data that better aligns with real-world applications. Unlike previous studies, we employed heuristic rules to create adversarial LLM-generated text, simulating various prompts usages, human revisions like word substitutions, and writing noises like spelling mistakes. Our development of *DetectRL* reveals the strengths and limitations of current SOTA detectors. More importantly, we analyzed the potential impact of writing styles, model types, attack methods, the text lengths, and real-world human writing factors on different types of detectors. We believe *DetectRL* could serve as an effective benchmark for assessing detectors in real-world scenarios, evolving with advanced attack methods, thus providing more stressful evaluation to drive the development of more efficient detectors[2].

## 1   Introduction

Detecting text generated by LLMs is a challenging task. It is often more difficult for humans than for detection techniques to identify LLM-generated text, as humans typically underperform detection methods designed for this purpose [1]. Recently, the implications of LLM-generated content have come into focus, highlighting their significant societal and academic impacts and associated risks [2, 3]. The main concerns stem from the hallucinations and misuse of LLMs [4], leading to issues such as plagiarism [5], the spread of fake news [6], and challenges to educators and human scholarship in AI-assisted academic writing [7]. Previous and current popular detection benchmarks, such as TuringBench [8], MGTBench [9], MULTITuDE [10], MAGE [11] and M4 [12], have primarily focused on evaluating detectors' performance across various domains, generative models, and languages by constructing idealized test data. However, they have overlooked the assessment of detectors' capabilities in more common scenarios encountered in practical applications [4], such as various prompt usages, human revisions, and writing noises, as shown in Table 1.

In this paper, we study the following questions: **(1) How do SOTA LLM-generated text detectors perform in real-world application scenarios? (2) What real-world factors influence detector**

---

*Corresponding author
[2]Data and code are publicly available at: https://github.com/NLP2CT/DetectRL

Table 1: Comparison with existing benchmarks. ✓: benchmark evaluates this scenario. △: has studies, not in evaluation. ◯: similar scenario exist, but not fully aligns with real-world applications.

| Benchmark ↓ Eval → | Multi Domains | Multi LLMs | Various Prompts | Human Revision | Writing Noises | Data Mixing | Detector Generalization | Training Length | Test Length | Real World Human Writing |
|---|---|---|---|---|---|---|---|---|---|---|
| TuringBench [8] | ✓ | ✓ | - | - | - | - | - | - | - | - |
| MGTBench [9] | ✓ | ✓ | - | ◯ | ◯ | - | △ | - | △ | - |
| MULTITuDE [10] | ✓ | ✓ | - | - | - | - | △ | - | - | - |
| M4 [12] | ✓ | ✓ | ✓ | - | - | - | ✓ | - | - | - |
| MAGE [11] | ✓ | ✓ | - | ◯ | - | - | ✓ | - | - | - |
| *DetectRL* (Ours) | ✓ | ✓ | ✓ | ✓ | ✓ | ✓ | ✓ | ✓ | ✓ | ✓ |

**performance, and to what extent?** We investigate these questions by introducing *DetectRL*, a novel benchmark for LLM-generated text detection. We achieve this by crafting challenges that are commonly encountered in real-world scenarios. These challenges simulate various prompts usages of human, human revisions of text such as word substitutions, and adversarial writing noises, including spelling mistakes. To enhance these simulations, we incorporate well-designed attack methods like prompt-based attacks, paraphrasing, adversarial perturbations, and data mixing. We selected data from domains where LLMs are frequently used and prone to abuse, such as academic writing, news writing, creative writing, and social media, to serve as samples of human-written text. To create LLM-generated texts that closely resemble real-world application scenarios, we employed powerful and widely used LLMs, including GPT-3.5-turbo [13], PaLM-2-bison [14], Claude-instant [15], and Llama-2-70b [16]. Furthermore, to ensure a wider diversity of text length, we filtered out shorter texts and applied a varying length augmentation method. This approach significantly broadened the range of text lengths available for detection, enhancing the practical value of the task. We balanced sample distributions across domains, LLMs, and attack types in all test scenarios to enhance diversity, thereby creating more challenging evaluations. These distribution variances are common in real-world scenarios but are often overlooked in ideal test environments where current detectors are developed.

The construction of this benchmark was highly effective in achieving our goals. **The experimental results present a significant challenge to existing detection methods.** Current detectors, particularly those employing zero-shot techniques, often struggle with accurately identifying LLM-generated texts. For example, adversarial perturbation attacks reduce the performance of all zero-shot detectors by an average of 39.28% AUROC. In contrast, supervised detectors have demonstrated robust detection capabilities in various domains, generative models, and attacks settings.

Through our benchmark analysis, **we highlight the strong relationship between various factors and detector performance**. Key elements that undermine the robustness and generalization of detectors include the informal style of domain data, distinct statistical patterns of LLMs, and adversarial perturbation attacks. Our findings indicate that shorter training data is beneficial for building robust detectors, while longer test data improves detector performance. Additionally, when human-written text undergoes attacks, the impact on detector performance is minimal, and performance may even improve after perturbation. This underscores the potential for adversarial perturbations to enhance current detection capabilities. Furthermore, our proposed framework aims to support the long-term development of attack methods against detectors. This will enable the creation of more challenging benchmarks that reflect real-world usages and evaluate the effectiveness of detection methods.

## 2 *DetectRL*

Previous datasets were mainly constructed by directly collecting human-written texts and those generated by LLMs using the same questions or prompt prefixes. This approach assumes an ideal detection environment and overlooks critical design considerations such as application domains, generative models, potential attacks, and text lengths. We improve the current dataset construction approach to better align with real-world detection scenarios. In this section, we introduce *DetectRL*, a new benchmark designed to facilitate such assessments, with its overall framework shown in Figure 1.

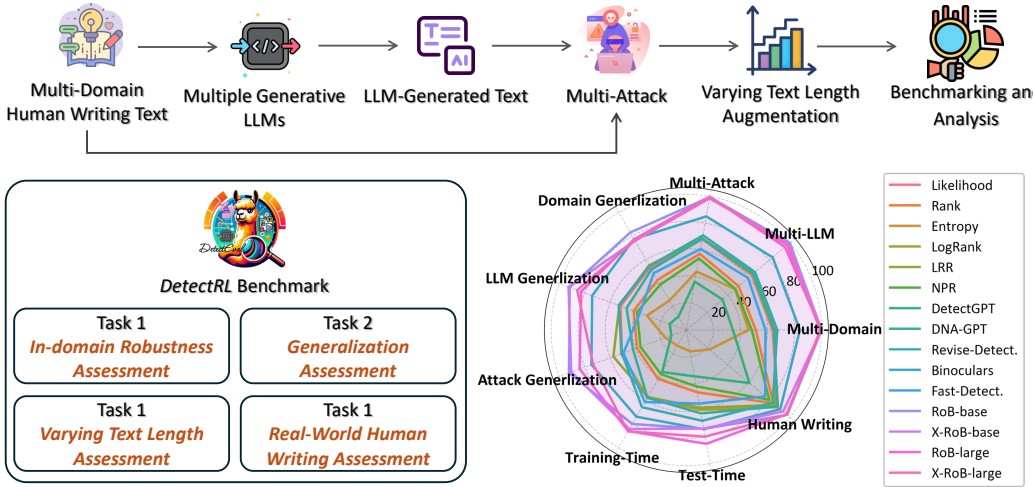

Figure 1: The overall framework of *DetectRL*. Human-written samples are collected from high-risk and abuse-prone domains. We employ widely-used and powerful LLMs to create LLM-generated samples. All samples undergo well-designed attacks to simulate real-world scenarios and a varying length augmentation method is applied to enhance the benchmark's diversity. *DetectRL* consists of four distinct tasks to evaluate the detectors' comprehensive detection abilities and robustness.

## 2.1 Framework

**Data sources** *DetectRL* is a comprehensive benchmark consisting of academic abstracts from the arXiv Archive,[3] covering the years 2002 to 2017. It also includes news articles from the XSum dataset [17], creative stories from Writing Prompts [18], and social reviews from Yelp Reviews [19]. The texts generated by LLMs within these domains are considered to pose higher risk of misleading content when misused, which underscores the importance of effective detection strategies. We extracted 2,800 samples per dataset as human-written texts. To avoid the potential contamination from text generated by LLMs, all selected data was released prior to the advent of ChatGPT.

**Models** Based on the collected human-written texts, we selected several LLMs that widely used in real-world, including GPT-3.5-turbo [13], PaLM-2-bison [14], Claude-instant [15], and Llama-2-70b [16], to perform text generation tasks. These models are mostly black-box and require substantial computational resources, making white-box detection methods challenging. We obtain text samples generated by these LLMs through interactive sessions with each model. For more details on the LLMs and text generation settings, please refer to Appendix D.

**Data generation** We employed various attack methods to simulate complex real-world detection scenarios. Following the classifications from the studies by [4] and [20], we categorized our attack methods into prompt attacks, paraphrase attacks, and perturbation attacks. Additionally, we treated data mixing as a separate scenario in our study. Please see Appendix D.4 for implementation details.

*Prompt attacks* are intended to use carefully designed prompts to guide LLMs in generating text that closely mimics human writing style. Our employed prompt attacks include Few-shot Prompting [21] and ICO Prompting, which is part of SICO Prompting [22].

*Paraphrase attacks* have been extensively studied in recent research on LLM-generated text detection [23], focusing on rewriting text while maintaining its original meaning. Alongside using the DIPPER-paraphraser [23], we also employed Back-translation via Google Translate[4] and Polishing using LLMs, which are two paraphrasing methods commonly utilized in everyday scenarios.

*Perturbation attacks* mainly involve introducing adversarial perturbations on text directly generated by LLMs. These attacks can effectively simulate common writing errors, character or word substitutions

---

[3]https://www.kaggle.com/datasets/spsayakpaul/arxiv-paper-abstracts/data
[4]https://translate.google.com/

or other adversarial noises in real-world applications. We utilized DeepWordBug [24] for Character-level Perturbations, TextFooler [25] for Word-level Perturbations, and TextBugger [24] for Sentence-level Perturbations, all implemented using TextAttack [26].

*Data Mixing* involves two primary approaches: Multi-LLM Mixing and LLM-Centered Mixing. In Multi-LLM Mixing, we create LLM-generated samples by sampling and combining sentences from multiple LLMs. On the other hand, LLM-Centered Mixing involves substituting one-fourth of an LLM-generated text with randomly selected human-written content. Despite this substitution, the text remains labeled as LLM-generated, since the majority originates from the LLM.

**Data augmentation**   We enhance the diversity of samples of different lengths through data augmentation, primarily by splitting texts at the sentence level. This approach creates multiple versions of each text sample with varying lengths. Based on the distribution of text lengths, we then categorize these sample into intervals of 20 words each (up to 360 words, since longer texts are rare). Within each interval, we uniformly sample 900 examples to comprehensively assess the detector's performance.

Table 2: Benchmark statistics.

| Task | Setting | Sub Setting | Training | | Test |
| --- | --- | --- | --- | --- | --- |
| | | | Supervised | Zero-Shot | |
| **Task 1** | Multi-Domain | Academic | 25,990 | 2,008 | 2,008 |
| | | News | 25,992 | 2,008 | 2,008 |
| | | Creative | 25,985 | 2,008 | 2,008 |
| | | Social Media | 25,984 | 2,008 | 2,008 |
| | Multi-LLM | GPT-3.5-turbo | 25,987 | 2,008 | 2,008 |
| | | Claude-instant | 25,990 | 2,008 | 2,008 |
| | | PaLM-2-bison | 25,987 | 2,008 | 2,008 |
| | | Llama-2-70b | 25,987 | 2,008 | 2,008 |
| | Multi-Attack | Direct | 20,384 | 2,016 | 2,016 |
| | | Prompt | 31,568 | 2,032 | 2,032 |
| | | Paraphrase | 42,767 | 2,016 | 2,016 |
| | | Perturbation | 42,784 | 2,016 | 2,016 |
| | | Data Mixing | 401,184 | 2,008 | 2,008 |
| **Task 2** | Domain Generalization | Academic | 25,990 | 2,008 | 6,024 |
| | | News | 25,992 | 2,008 | 6,024 |
| | | Creative | 25,985 | 2,008 | 6,024 |
| | | Social Media | 25,984 | 2,008 | 6,024 |
| | LLM Generalization | GPT-3.5-turbo | 25,987 | 2,008 | 6,024 |
| | | Claude-instant | 25,990 | 2,008 | 6,024 |
| | | PaLM-2-bison | 25,987 | 2,008 | 6,024 |
| | | Llama-2-70b | 25,987 | 2,008 | 6,024 |
| | Attack Generalization | Direct | 20,384 | 2,016 | 6,048 |
| | | Prompt | 31,568 | 2,032 | 6,096 |
| | | Paraphrase | 42,767 | 2,016 | 6,048 |
| | | Perturbation | 42,784 | 2,016 | 6,048 |
| | | Data Mixing | 401,184 | 2,008 | 6,024 |
| **Task 3** | Varying Text Length | Training-Time | 16,200 | 16,200 | 900 |
| | | Test-Time | 900 | 900 | 16,200 |
| **Task 4** | Human Writing | Direct | 20,384 | 2,016 | 2,016 |
| | | Paraphrase | 42,767 | 2,016 | 2,016 |
| | | Perturbation | 42,784 | 2,016 | 2,016 |
| | | Data Mixing | 42,788 | 2,012 | 2,012 |

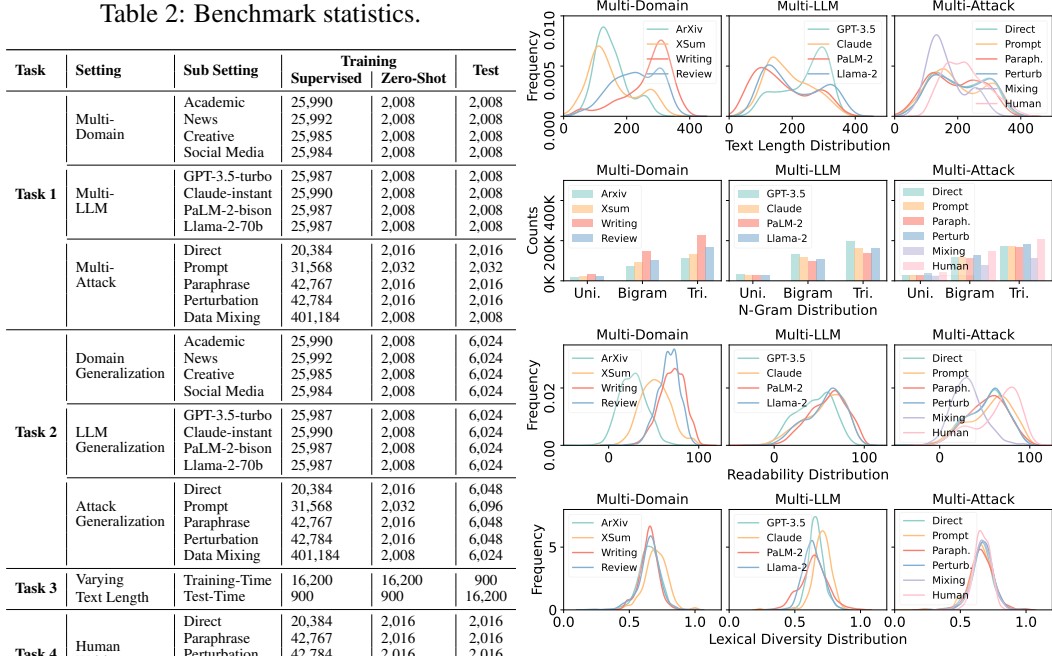

Figure 2: Benchmark textual features statistics.

## 2.2  Task definition

Based on the meticulously curated dataset, we manifest the ***DetectRL*** framework into four distinct tasks for LLM-generated text detectors assessment, described as follows:

**Task 1: In-domain robustness assessment: multi-domain, multi-LLM, and multi-attack assessment.**   This task aims to evaluate the foundational performance of detectors in different domains, generators, and attack strategies, focusing specifically on their in-domain robustness in various real-world scenarios. We use the average performance score as the assessment metric.

**Task 2: Generalization assessment.**   This task assesses the generalization of detectors from three perspectives: domain, LLM, and attack, to determine their effectiveness in diverse scenarios. Unlike Task 1, this task emphasizes the detector's ability to handle out-of-distribution samples. For example, we evaluate the performance of detectors trained on texts from one domain when applied to texts from different domains to determine their generalization score across domains. The same approach is used to assess generalization across different LLMs and attack strategies.

**Task 3: Varying text length assessment.** This task evaluates the impact of text length on the performance of detectors, considering both training-time and test-time phase. In the training-time phase, detectors are trained on samples of different length intervals and then tested on samples from the pivotal interval. In the test-time phase, the detector trained on samples from the pivotal interval is evaluated with samples of varying lengths. This approach provides a comprehensive understanding of how text length influences detection capabilities.

**Task 4: Real-world human writing assessment.** This task evaluates how real-world human writing factors impact the performance of detectors. In this innovative assessment, we simulate and replicate these factors like word substitutions and spelling errors by applying attacks on human-written texts, highlighting the challenges they pose to detectors in real-world scenarios.

## 2.3 Benchmark statistics

The statistics for the collected data are presented in Appendix Table 9. This dataset comprises 100,800 human-written samples, including 11,200 raw samples and 89,600 samples modified via attack manipulations. Additionally, it contains 134,400 samples generated by LLMs, categorized as follows: 11,200 samples generated with direct prompt, 22,400 with prompt attacks, 33,600 with paraphrase attacks, 33,600 with perturbation attacks, and 22,400 with data mixing. To evaluate detectors performance, we designed the **DetectRL** benchmark by carefully extracting relevant subsets of data to align with the task design. The selected samples ensure a balance across domains, LLMs, and attack types. The training data was specifically tailored for both supervised and zero-shot detectors, and performance was evaluated using common test sets. Detailed statistics for each task and the analysis of the textual features of **DetectRL** samples are presented in Figure 2. For a more detailed analysis, please refer to Appendix D.6.

## 2.4 Evaluation metrics

We employ AUROC and $F_1$ Score as the main evaluation metrics. AUROC is widely used for assessing zero-shot detection methods [27] because it considers the True Positive Rate (TPR) and False Positive Rate (FPR) across different classification thresholds. This makes AUROC particularly useful for evaluating detector performance at different thresholds. The $F_1$ Score provides a comprehensive evaluation of detector capabilities by balancing the Precision and Recall. Additionally, we provide detailed Precision and Recall scores in Appendix F for further reference, with a specific focus on Recall to highlight the detectors' effectiveness in identifying LLM-generated text.

# 3 Experiments and discussion

In this section, we organize our experiments and discussions from five distinct perspectives: **(1) Benchmarking the cutting-edge detectors**: We evaluate the current SOTA detectors against our benchmark to identify ongoing challenges. **(2) Robustness analysis**: We analyze the factors contributing to robustness issues across various domains, LLMs, and attack scenarios. **(3) Assessing generalization**: We investigate how well detectors perform on data distribution they were not specifically trained on, highlighting their out-of-distribution robustness. **(4) Length discrimination**: We examine the detectors' ability to differentiate between texts of varying lengths and discuss the impact of training on such texts. **(5) Real-world human writing scenarios**: We analyze the effects of real-world post-processing and mistake in human-written texts, discussing their implications to provide more nuanced and valuable insights.

## 3.1 Benchmarking detectors

**Detectors** We employed a variety of SOTA detectors to assess the difficulty of **DetectRL**. Given that LLMs in real-world scenarios are often black-box and inaccessible, we exclude watermarking methods from our evaluation. Our evaluation encompasses prominent zero-shot techniques and supervised fine-tuned classifiers, including Log-Likelihood [28], Entropy [29], Rank [30], Log-Rank [30], LRR [31], NPR [31], DetectGPT [27], Fast-DetectGPT [32], Revise-Detect. [33], DNA-GPT [34], Binoculars [35], RoBERTa Classifier (RoB [36]), and XLM-RoBERTa Classifier[5] (X-RoB [37]).

---

[5]Please refer to Appendix E.2 for comprehensive detectors specifications.

For the white-box zero-shot detection method, we employ the GPT-Neo-2.7B [38] as the scoring model, in line with the methodology proposed in Fast-DetectGPT [32], to detect the text generated by black-box LLMs. For the black-box zero-shot detection method like Revise-Detect. [33] and DNA-GPT [34], we use GPT-4o-Mini [39] to perform operations such as text revision and text continuation. For supervised detectors, all classifiers are trained using the same parameters. For detailed training parameters settings, please refer to Table 12 and AppendixE.2.

Table 3: The overall leaderboard for LLM-generated text detectors in real-world scenarios ranks detectors based on their robustness and generalization across various domains, LLMs, and attack scenarios. It also considers the impact of text length in training-time and test-time phase, as well as performance against real-world human writing factors.

| Leaderboard: LLM-Generated Text Detector in Real-World Scenarios | | | | | | | | | | | | | |
|---|---|---|---|---|---|---|---|---|---|---|---|---|---|
| Tasks Settings → | Multi-Domain | | Multi-LLM | | Multi-Attack | | Generalization | | | Time | | Human Writing | | Avg. |
| Detectors ↓ | AUROC | $F_1$ | AUROC | $F_1$ | AUROC | $F_1$ | Domain $F_1$ | LLM $F_1$ | Attack $F_1$ | Train $F_1$ | Test $F_1$ | AUROC | $F_1$ | |
| Rob-Base | 99.98 | 99.75 | 99.93 | 99.58 | 99.56 | 97.66 | 83.00 | 91.81 | 92.37 | 79.99 | 74.00 | 97.34 | 94.31 | 93.02 |
| Rob-Large | 99.78 | 98.87 | 95.16 | 90.03 | 99.87 | 99.03 | 77.20 | 82.85 | 83.96 | 86.08 | 85.23 | 96.68 | 94.63 | 91.49 |
| X-Rob-Base | 99.92 | 99.34 | 99.14 | 98.17 | 98.49 | 96.07 | 75.97 | 92.73 | 90.58 | 84.25 | 73.83 | 93.43 | 90.29 | 91.71 |
| X-Rob-Large | 99.01 | 97.44 | 97.40 | 93.47 | 99.31 | 97.75 | 76.14 | 85.89 | 73.42 | 86.35 | 79.83 | 97.21 | 94.43 | 90.59 |
| Binoculars | 83.95 | 78.25 | 83.30 | 74.83 | 85.05 | 78.53 | 77.47 | 74.10 | 74.70 | 73.82 | 74.34 | 90.68 | 85.98 | 79.61 |
| Revise-Detect. | 67.24 | 60.82 | 66.36 | 53.72 | 70.89 | 57.24 | 54.50 | 53.28 | 50.63 | 65.71 | 67.96 | 83.29 | 82.16 | 64.13 |
| Log-Rank | 64.43 | 57.53 | 63.75 | 54.18 | 68.52 | 55.15 | 55.10 | 52.78 | 51.28 | 57.44 | 59.74 | 88.46 | 83.85 | 62.48 |
| LRR | 65.47 | 55.45 | 64.93 | 53.01 | 68.53 | 57.99 | 54.61 | 52.73 | 57.41 | 57.09 | 58.15 | 85.99 | 80.56 | 62.46 |
| Log-Likelihood | 63.71 | 56.36 | 62.97 | 53.13 | 67.97 | 54.38 | 53.37 | 51.77 | 50.73 | 57.92 | 59.28 | 88.48 | 83.75 | 61.83 |
| DNA-GPT | 64.92 | 55.83 | 64.36 | 51.09 | 68.36 | 53.36 | 51.51 | 47.09 | 41.98 | 57.63 | 62.43 | 87.80 | 82.77 | 60.70 |
| Fast-DetectGPT | 58.52 | 48.07 | 59.58 | 46.55 | 60.70 | 50.63 | 48.35 | 36.56 | 49.47 | 61.31 | 55.08 | 76.03 | 68.47 | 55.33 |
| Rank | 51.34 | 44.97 | 50.33 | 42.06 | 57.08 | 48.83 | 42.61 | 41.49 | 38.84 | 41.67 | 46.65 | 83.86 | 80.00 | 51.52 |
| NPR | 48.37 | 41.41 | 47.27 | 40.04 | 53.49 | 45.22 | 38.58 | 38.83 | 36.10 | 37.60 | 42.17 | 80.03 | 75.98 | 48.08 |
| DetectGPT | 34.43 | 21.52 | 34.93 | 14.80 | 36.19 | 19.15 | 11.54 | 13.11 | 11.84 | 35.78 | 34.69 | 60.86 | 48.76 | 29.05 |
| Entropy | 46.02 | 27.40 | 46.97 | 34.25 | 43.75 | 24.69 | 25.06 | 31.07 | 16.53 | 13.38 | 15.99 | 22.39 | 16.60 | 28.01 |

**Main results**  We assessed the performance of existing detectors on *DetectRL*, as shown in Table 3. Higher average scores indicate greater utility of the detector. These results highlight the challenges posed by our benchmark and explain why current SOTA detectors have not been widely adopted. The leaderboard results demonstrate that supervised detectors consistently outperform zero-shot detectors, demonstrating greater effectiveness and robustness. Among the zero-shot methods, Binoculars ranked highest but scored only 79.61%. The second-best is Revise-Detect., scoring 64.13%, followed by Log-Rank, LRR, Log-Likelihood, DNA-GPT, and Fast-DetectGPT. Additionally, our analysis highlights the unreliability of advanced detectors such as DetectGPT and NPR in real-world applications.

**Significant Challenges**  Our benchmarks reveal significant challenges in the current LLM-generated text detection research. We found that incorporating a mix distribution of domains, LLMs, and attack types increases the testing pressure of zero-shot methods. For example, in the multi-LLM setting, the average AUROC of all zero-shot detectors is only 58.61%. This is because data from each LLM spans various domains and attack methods, leading to substantial distribution differences even within data from the same LLM. These variations are often overlooked in ideal testing environments, making it difficult for zero-shot detectors developed based on them to work effectively. Specifically, zero-shot detectors struggle against powerful LLMs, achieving an average AUROC of only 77.67% on texts generated via direct prompting, with only Binoculars surpassing a 90% AUROC. The performance of these detectors declines markedly under well-designed attacks that simulate real-world scenarios, with average decreases of 1.97% in prompt attacks, 15.67% in paraphrase attacks, 38.43% in perturbation attacks, and 18.17% in data mixing scenarios. In contrast, supervised methods demonstrate impressive effectiveness, achieving an average AUROC of 99.40% on data generated through direct prompting and maintaining robustness against well-designed attacks.

Unexpectedly, recent advancements in LLM-generated text detection, such as DetectGPT [27], NPR [31], Fast-DetectGPT [32], and DNA-GPT [34], did not perform as expected on our benchmark. Their performance was even weaker than some traditional zero-shot baselines. Analysis across various domains and LLMs revealed a general lack of robustness as a potential underlying issue. For instance, DetectGPT's performance was notably low, with only 22.15% AUROC in academic writing (ArXiv) and 12.21% AUROC in news writing (Xsum), though it achieved 58.95% AUROC in creative writing and 44.43% AUROC in social media (Yelp Review). A similar trend was observed with the

Table 4: The performance of detectors in multi-domain, multi-LLM, and multi-attack assessment. The shades of blue and red illustrate the performance differences between the zero-shot and the supervised detectors, respectively. The underlined values represent the best performance.

**Multi-Domain**

| Metrics → | - | ArXiv AUROC | ArXiv $F_1$ | XSum AUROC | XSum $F_1$ | Writing AUROC | Writing $F_1$ | Review AUROC | Review $F_1$ | Avg. AUROC | Avg. $F_1$ |
|---|---|---|---|---|---|---|---|---|---|---|---|
| Log-Likelihood | - | 65.35 | 57.55 | 45.68 | 41.32 | 68.00 | 59.38 | 75.84 | 67.22 | 63.7122 | 56.367 |
| Entropy | - | 48.39 | 29.71 | 67.84 | 57.23 | 39.06 | 20.55 | 28.82 | 02.14 | 46.0253 | 27.4066 |
| Rank | - | 57.17 | 54.62 | 36.87 | 22.47 | 56.26 | 50.90 | 55.08 | 51.90 | 51.3409 | 44.97 |
| Log-Rank | - | 67.01 | 60.09 | 46.74 | 42.60 | 67.58 | 57.57 | 76.40 | 69.88 | 64.43 | 57.5378 |
| LRR | - | 70.54 | 61.34 | 50.09 | 38.38 | 64.65 | 53.09 | 76.61 | 68.99 | 65.47 | 55.4570 |
| NPR | - | 53.85 | 49.65 | 34.59 | 18.31 | 54.96 | 52.30 | 50.09 | 45.39 | 48.387 | 41.416 |
| DetectGPT | - | 22.15 | 00.00 | 12.21 | 00.00 | 58.95 | 50.83 | 44.43 | 35.25 | 34.434 | 21.502 |
| DNA-GPT | - | 67.41 | 58.30 | 64.22 | 45.09 | 69.04 | 58.25 | 78.17 | 69.28 | 69.71 | 57.723 |
| Revise-Detect. | - | 70.40 | 37.51 | 50.34 | 46.07 | 73.24 | 64.29 | 75.01 | 68.71 | 67.2475 | 54.1465 |
| Binoculars | - | 84.03 | 76.77 | 77.39 | 72.18 | 94.38 | 79.73 | 90.00 | 84.32 | 86.95 | 78.75 |
| Fast-DetectGPT | - | 43.69 | 24.46 | 39.19 | 28.39 | 74.21 | 67.84 | 77.02 | 71.62 | 58.03 | 48.08 |
| **Avg.** | - | 59.09 | 46.36 | 47.374 | 37.45 | 65.48 | 55.88 | 66.13 | 57.70 | 59.68 | 49.39 |
| Rob-Base | - | 100.0 | 100.0 | 99.99 | 99.85 | 99.99 | 99.65 | 99.97 | 99.50 | 99.99 | 99.75 |
| Rob-Large | - | 99.99 | 99.90 | 99.85 | 98.95 | 99.54 | 97.73 | 99.76 | 98.90 | 99.54 | 98.87 |
| X-Rob-Base | - | 100.0 | 100.0 | 99.97 | 99.55 | 99.84 | 98.76 | 99.88 | 99.05 | 99.92 | 99.59 |
| X-Rob-Large | - | 99.98 | 99.85 | 99.84 | 98.95 | 99.85 | 98.31 | 96.40 | 92.66 | 99.23 | 97.19 |
| **Avg.** | - | 99.99 | 99.93 | 99.91 | 99.32 | 99.80 | 98.61 | 99.00 | 97.52 | 99.67 | 98.85 |

**Multi-LLM**

| LLM Settings → | - | GPT-3.5 AUROC | GPT-3.5 $F_1$ | Claude AUROC | Claude $F_1$ | PaLM-2 AUROC | PaLM-2 $F_1$ | Llama-2 AUROC | Llama-2 $F_1$ | Avg. AUROC | Avg. $F_1$ |
|---|---|---|---|---|---|---|---|---|---|---|---|
| Log-Likelihood | - | 62.89 | 57.80 | 43.32 | 28.10 | 70.03 | 60.73 | 75.65 | 65.90 | 62.97 | 53.13 |
| Entropy | - | 46.84 | 23.29 | 52.25 | 30.42 | 45.34 | 16.56 | 43.48 | 66.75 | 46.97 | 34.25 |
| Rank | - | 52.19 | 49.32 | 41.68 | 22.78 | 50.40 | 41.74 | 57.05 | 54.40 | 50.33 | 42.06 |
| Log-Rank | - | 62.84 | 56.87 | 43.32 | 30.12 | 70.89 | 63.09 | 77.97 | 66.66 | 63.75 | 54.18 |
| LRR | - | 61.61 | 52.12 | 43.30 | 18.91 | 71.17 | 65.51 | 83.65 | 75.51 | 64.93 | 53.01 |
| NPR | - | 50.29 | 43.81 | 41.64 | 32.91 | 44.64 | 34.77 | 52.53 | 48.68 | 47.27 | 40.04 |
| DetectGPT | - | 43.46 | 26.27 | 32.86 | 12.56 | 26.72 | 00.00 | 36.71 | 20.40 | 34.93 | 14.80 |
| DNA-GPT | - | 61.87 | 55.04 | 48.88 | 25.67 | 71.48 | 60.77 | 75.22 | 62.89 | 64.36 | 51.09 |
| Revise-Detect. | - | 70.10 | 62.72 | 49.87 | 27.28 | 69.84 | 59.03 | 75.65 | 65.87 | 66.36 | 53.72 |
| Binoculars | - | 88.14 | 82.50 | 55.15 | 39.35 | 93.30 | 88.20 | 96.64 | 92.30 | 83.30 | 75.58 |
| Fast-DetectGPT | - | 65.56 | 59.55 | 30.01 | 00.00 | 65.99 | 57.58 | 76.79 | 69.08 | 59.58 | 46.55 |
| **Avg.** | - | 60.52 | 51.75 | 43.84 | 24.37 | 61.80 | 49.81 | 68.30 | 62.58 | 58.61 | 47.12 |
| Rob-Base | - | 99.97 | 99.70 | 99.98 | 99.80 | 99.94 | 99.40 | 99.84 | 99.45 | 99.93 | 99.59 |
| Rob-Large | - | 99.77 | 98.86 | 96.23 | 92.48 | 97.93 | 92.64 | 86.72 | 76.17 | 95.66 | 90.54 |
| X-Rob-Base | - | 99.88 | 99.45 | 98.26 | 97.48 | 98.77 | 97.19 | 99.69 | 98.57 | 99.15 | 98.17 |
| X-Rob-Large | - | 99.55 | 97.56 | 91.67 | 84.24 | 98.73 | 94.43 | 99.66 | 97.67 | 97.65 | 93.73 |
| **Avg.** | - | 99.79 | 98.89 | 96.53 | 93.50 | 98.84 | 95.91 | 96.47 | 92.96 | 98.09 | 95.50 |

**Multi Attack**

| Attack Settings → | Direct AUROC | Direct $F_1$ | Prompt AUROC | Prompt $F_1$ | Paraph. AUROC | Paraph. $F_1$ | Perturb AUROC | Perturb $F_1$ | Mixing AUROC | Mixing $F_1$ | Avg. AUROC | Avg. $F_1$ |
|---|---|---|---|---|---|---|---|---|---|---|---|---|
| Log-Likelihood | 89.25 | 82.09 | 86.87 | 78.16 | 64.55 | 57.59 | 35.51 | 00.78 | 63.70 | 53.31 | 67.97 | 54.38 |
| Entropy | 26.47 | 00.00 | 26.18 | 00.00 | 48.12 | 26.01 | 68.62 | 68.95 | 49.37 | 28.52 | 43.75 | 24.69 |
| Rank | 83.50 | 76.27 | 81.21 | 72.86 | 60.60 | 52.60 | 08.04 | 00.00 | 52.05 | 42.46 | 57.08 | 48.83 |
| Log-Rank | 89.25 | 81.45 | 86.35 | 77.51 | 64.69 | 59.17 | 37.71 | 00.78 | 64.63 | 56.86 | 68.52 | 55.15 |
| LRR | 85.83 | 77.40 | 80.80 | 74.30 | 63.99 | 55.20 | 45.91 | 29.27 | 66.12 | 53.81 | 68.53 | 57.99 |
| NPR | 77.98 | 71.61 | 77.15 | 70.63 | 56.94 | 46.25 | 06.78 | 00.00 | 48.63 | 37.65 | 53.49 | 45.22 |
| DetectGPT | 52.84 | 40.90 | 51.83 | 37.98 | 31.79 | 16.89 | 18.21 | 00.00 | 26.28 | 00.00 | 36.19 | 19.15 |
| DNA-GPT | 88.01 | 80.78 | 85.62 | 77.47 | 65.61 | 54.94 | 40.45 | 02.73 | 62.14 | 50.89 | 68.77 | 53.76 |
| Revise-Detect. | 86.88 | 79.61 | 84.89 | 76.21 | 67.26 | 62.03 | 43.98 | 07.56 | 65.27 | 54.39 | 69.26 | 56.76 |
| Binoculars | 94.87 | 89.73 | 93.45 | 88.12 | 88.34 | 81.56 | 76.89 | 69.34 | 89.12 | 83.67 | 88.53 | 82.48 |
| Fast-DetectGPT | 79.56 | 72.45 | 78.43 | 70.34 | 70.12 | 62.89 | 49.56 | 41.23 | 67.23 | 59.78 | 68.58 | 61.34 |
| **Avg.** | 77.67 | 68.39 | 75.70 | 65.78 | 62.00 | 52.28 | 39.24 | 20.05 | 59.50 | 47.39 | 62.78 | 50.88 |
| Rob-Base | 99.87 | 99.60 | 99.78 | 99.47 | 99.67 | 99.12 | 98.32 | 97.45 | 99.12 | 98.76 | 99.35 | 98.88 |
| Rob-Large | 98.73 | 97.83 | 98.45 | 97.56 | 97.89 | 96.78 | 96.12 | 94.67 | 97.56 | 96.34 | 97.75 | 96.64 |
| X-Rob-Base | 99.56 | 99.12 | 99.23 | 99.01 | 98.89 | 98.34 | 98.56 | 97.89 | 99.01 | 98.56 | 98.85 | 98.58 |
| X-Rob-Large | 99.45 | 98.67 | 98.89 | 97.98 | 98.23 | 97.67 | 97.89 | 96.34 | 98.67 | 97.89 | 98.63 | 97.71 |
| **Avg.** | 99.40 | 98.80 | 99.09 | 98.50 | 98.67 | 97.98 | 97.22 | 96.09 | 98.34 | 97.89 | 98.54 | 97.85 |

best zero-shot detector, Binoculars, which performed more than 10% lower in academic writing and news writing compared to other domains. Additionally, Binoculars showed significantly reduced effectiveness on text generated by Claude, achieving only 55.15% AUROC, while presenting 88.14%,

93.30%, and 96.64% AUROC on text generated by GPT-3.5, PaLM-2, and Llama-2, respectively. These findings suggest that the performance differences of detectors across different domains and LLMs become significantly more pronounced when subjected to well-designed attacks.

## 3.2 In-domain Robustness

**Effectiveness of zero-shot detectors varies with the stylistic nature of domain data.** As shown in Table 4, our results indicate that texts with a more formal style present greater challenges for detection. Detectors generally perform better with informal data, such as that from social media, but their effectiveness decreases markedly in more formal settings like news writing. Interestingly, this decrease in performance is even more pronounced in advanced detectors like Fast-DetectGPT [32]. Despite this variability, supervised classifiers demonstrate consistent reliability in detection across various domains. This finding aligns with insights from [40], emphasizing the robustness of supervised classifiers in diverse textual environments.

**Differences in statistical patterns of LLMs pose significant challenges to detectors.** As illustrated in Table 4, our experiments reveal a notable phenomenon: nearly all zero-shot LLM-generated text detectors exhibit a significant decline in performance when processing texts generated by Claude. This suggests that the effectiveness of detectors is influenced by the type of generative model used to generate the text to be detected, and their performance can deteriorate with varying statistical patterns. We hypothesize that these differences arise from variations in data, architecture, and training methods of the models, though verifying this is difficult due to the opaque nature of black-box models. Moreover, supervised detectors are more affected by the type of generative model than by the domain, particularly in models with larger sizes. For example, Rob-Large achieved an AUROC of only 86.72% and an $F_1$ Score of only 76.17% on texts generated by Llama-2, while X-Rob-Large achieved an AUROC of only 91.67% and an $F_1$ Score of only 82.24% on texts generated by Claude.

**Adversarial perturbation attacks represent a significant threat to zero-shot detectors.** As shown in Table 4, our findings indicate that the adversarial perturbation attacks drastically reduce the effectiveness of zero-shot detectors, reducing their performance to an average AUROC of 38.43%, which is less than half compared to their performance under paraphrase attacks. Additionally, data mixing presents a new challenging scenario, resulting in performance levels similar to paraphrase attacks, with detectors achieving an average AUROC of 59.50%. While prompt attacks, such as few-shot prompting, can generate higher-quality text more aligned with human preferences, their impact on zero-shot detectors is minimal. However, enhancing LLM-generated texts through human-written prompts, such as those used for polishing, continues to pose challenges for detectors (see Appendix F.1), decreasing their effectiveness by an average of 8.97% AUROC. This finding suggests that prompt-based methods remain a viable means of compromising detector performance. In contrast, supervised detectors consistently maintain robust performance across various attack types, demonstrating their potential for practical applications.

## 3.3 Generalization of detectors

In real-world applications, there is a significant demand for detectors that can effectively adapt to various types of text. In this paper, we further investigate this requirement, specifically focusing on the relationship between the distribution of training and test data for these detectors. We assessed the generalization of three representative detectors: LRR [31], Fast-DetectGPT [32], and the RoB-Base Classifier [36]. We discussed their generalization from three perspectives: domain, LLM, and attack. Notably, we observed phenomena that align with the findings discussed in Section 3.2.

As shown in Table 5, our experimental results indicate that detectors trained on less formal stylistic domain data, such as creative writing and social media, exhibit stronger generalization. Their comprehensive performance is around 10% AUROC better than detectors trained on more formal stylistic domain data, such as academic writing and news writing. The variations in statistical patterns of generative models significantly impact the generalization of detectors. Detectors trained on texts generated by models with similar statistical patterns, such as GPT-3.5, PaLM-2, and Llama-2, generally perform well with each other. However, they struggle with texts generated by Claude. As discussed in Section 3.2, data with perturbation attacks poses the greatest challenge for generalization. Taking LRR as an example, the average AUROC for detectors trained on data with direct prompts,

Table 5: The performance of selected detectors in generalization assessment. The shades of blue and red illustrate the performance differences between the zero-shot and the supervised detectors.

| Detectors → | LRR (Zero-shot) | | | | | Fast-DetectGPT (Zero-shot) | | | | | Rob-Base (Supervised) | | | | |
|---|---|---|---|---|---|---|---|---|---|---|---|---|---|---|---|
| **Multi-Domain** | | | | | | | | | | | | | | | |
| Train ↓ Eval → | ArXiv | XSum | Writing | Review | Avg. | ArXiv | XSum | Writing | Review | Avg. | ArXiv | XSum | Writing | Review | Avg. |
| **ArXiv** | 57.55 | 40.88 | 38.44 | 55.81 | 48.17 | 24.46 | 23.71 | 59.67 | 60.17 | 42.00 | 100.0 | 75.90 | 77.68 | 70.69 | 81.06 |
| **XSum** | 57.45 | 41.32 | 39.08 | 55.81 | 48.41 | 28.43 | 28.39 | 62.99 | 63.08 | 45.72 | 68.43 | 99.85 | 71.79 | 67.17 | 76.81 |
| **Writing** | 61.14 | 46.31 | 59.38 | 67.98 | 58.70 | 34.81 | 33.60 | 67.84 | 68.30 | 51.13 | 78.58 | 72.72 | 99.65 | 94.24 | 86.29 |
| **Review** | 61.49 | 47.02 | 57.12 | 67.22 | 58.21 | 40.70 | 37.66 | 68.25 | 71.62 | 54.55 | 82.64 | 84.15 | 85.10 | 99.50 | 87.84 |
| **Multi-LLM** | | | | | | | | | | | | | | | |
| Train ↓ Eval → | GPT-3.5 | PaLM-2 | Claude | Llama-2 | Avg. | GPT-3.5 | PaLM-2 | Claude | Llama-2 | Avg. | GPT-3.5 | PaLM-2 | Claude | Llama-2 | Avg. |
| **GPT-3.5** | 52.12 | 61.79 | 24.70 | 75.34 | 53.48 | 59.55 | 59.56 | 12.96 | 69.93 | 50.50 | 99.97 | 70.34 | 62.90 | 94.68 | 81.97 |
| **PaLM-2** | 52.36 | 65.51 | 26.23 | 75.58 | 54.92 | 55.77 | 57.58 | 08.20 | 68.43 | 47.49 | 99.25 | 99.40 | 93.43 | 99.25 | 97.83 |
| **Claude** | 45.73 | 57.66 | 18.91 | 72.67 | 48.74 | 00.19 | 00.00 | 00.00 | 01.18 | 00.34 | 96.83 | 83.92 | 99.80 | 89.77 | 92.58 |
| **Llama-2** | 52.14 | 62.23 | 25.25 | 75.51 | 53.78 | 56.28 | 57.74 | 08.65 | 69.08 | 47.93 | 99.45 | 93.02 | 87.56 | 99.45 | 94.87 |
| **Multi-Attack** | | | | | | | | | | | | | | | |
| Train ↓ Eval → | Prompt | Paraph. | Perturb | Mixing | Avg. | Prompt | Paraph. | Perturb | Mixing | Avg. | Prompt | Paraph. | Perturb | Mixing | Avg. |
| **Direct** | 74.23 | 58.35 | 30.69 | 56.42 | 54.92 | 64.01 | 40.45 | 41.02 | 31.81 | 44.32 | 95.73 | 94.91 | 64.32 | 89.07 | 86.00 |
| **Prompt** | 74.30 | 58.35 | 30.81 | 56.42 | 54.97 | 64.00 | 39.94 | 40.40 | 31.25 | 43.89 | 97.18 | 94.98 | 86.18 | 92.92 | 92.81 |
| **Paraphrase** | 70.22 | 55.20 | 20.25 | 51.26 | 49.23 | 61.54 | 38.32 | 36.86 | 27.90 | 41.15 | 93.66 | 98.26 | 78.81 | 89.38 | 90.02 |
| **Perturb** | 71.81 | 58.22 | 29.27 | 55.19 | 53.62 | 64.01 | 40.45 | 41.14 | 31.93 | 44.38 | 87.01 | 91.46 | 98.66 | 91.38 | 92.12 |
| **Mixing** | 71.02 | 55.77 | 24.01 | 53.81 | 51.15 | 65.89 | 46.38 | 45.78 | 40.93 | 49.74 | 93.46 | 91.93 | 95.26 | 93.64 | 93.57 |

prompt attacks, paraphrase attacks, and data mixing, and then tested on data with perturbation attacks, is only 27.00%. This performance is 45.31%, 30.17%, and 27.62% lower than when tested on data with prompt attacks, paraphrase attacks, and data mixing, respectively. However, detectors trained on data with perturbation attacks do not show a significant decline in performance when tested on other types of attacks. This indicates that perturbation attack data may possess inherent complexities that are particularly challenging to detect.

## 3.4 Impact of text length

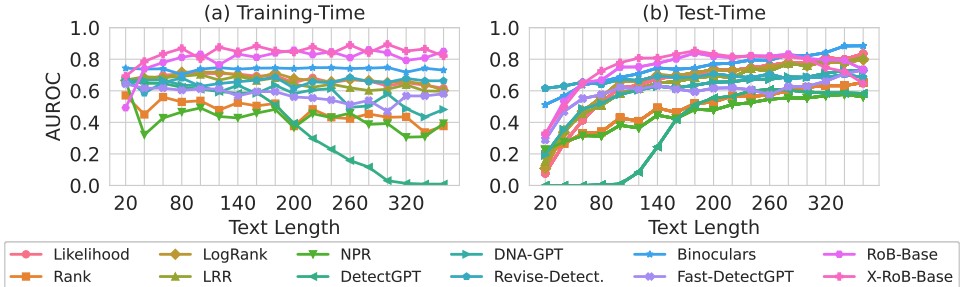

Figure 3: Impact of text length on AUROC during training-time and test-time.

**Shorter training samples for stronger detectors.** We assessed the performance of detectors trained on datasets with varying text lengths, using a test set within a specific pivot length interval of 160-180 words. The results, as shown in Figure 3 (a), revealed a golden length interval of 60-80 words, where texts consistently demonstrated strong detection performance across all detectors. However, as the length of the training texts increased, the performance of all zero-shot detectors gradually declined. This indicates that zero-shot detectors trained on shorter texts might be more effective than those trained on longer texts. In contrast, supervised detectors maintained consistent performance both within the golden length interval and in tests involving longer text lengths.

**Longer test samples for better zero-shot detection.** Similarly, we trained detectors using data from the pivotal length interval and assessed their performance on test sets with varying text lengths. The experimental results, shown in Figure 3 (b), reveal that as test text length increased, the performance of the zero-shot detectors improved steadily. This suggests a positive correlation between zero-

shot detectors' performance and test text length. In contrast, supervised methods showed a rapid performance increase up to the pivotal length interval, followed by a slight decline.

### 3.5  Impact of real-world human writing scenarios

Table 6: The performance of detectors in real-world human writing assessment. The shades of blue and red illustrate the performance differences between the zero-shot and the supervised detectors, respectively. The underlined values represent the best performance.

| Settings → | Direct | | Paraphrase Attack | | Perturbation Attack | | Data Mixing | | Avg. | |
|---|---|---|---|---|---|---|---|---|---|---|
| Detectors ↓ | AUROC | $F_1$ | AUROC | $F_1$ | AUROC | $F_1$ | AUROC | $F_1$ | AUROC | $F_1$ |
| **Zero-shot Detectors** | | | | | | | | | | |
| Log-Likelihood | 89.25 | 82.09 | 76.77 | 74.28 | 99.53 | 97.76 | 88.40 | 80.88 | 88.48 | 83.75 |
| Entropy | 26.47 | 00.00 | 27.15 | 00.00 | 03.37 | 00.00 | 32.58 | 66.40 | 22.39 | 16.60 |
| Rank | 83.50 | 76.27 | 72.14 | 74.13 | 99.63 | 98.13 | 80.17 | 71.48 | 83.86 | 80.00 |
| Log-Rank | 89.25 | 81.45 | 76.78 | 75.17 | 99.49 | 97.57 | 88.32 | 81.23 | 88.46 | 83.85 |
| LRR | 85.83 | 77.40 | 76.05 | 74.46 | 98.09 | 94.78 | 83.99 | 75.60 | 85.99 | 80.56 |
| NPR | 77.98 | 71.61 | 69.82 | 70.60 | 98.35 | 95.51 | 73.97 | 66.22 | 80.03 | 75.98 |
| DetectGPT | 52.84 | 40.90 | 68.45 | 73.45 | 87.95 | 79.74 | 34.20 | 00.98 | 60.86 | 48.76 |
| DNA-GPT | 88.01 | 80.78 | 77.19 | 75.95 | 98.81 | 95.83 | 87.40 | 76.55 | 87.85 | 82.27 |
| Revise-Detect. | 86.88 | 79.61 | 65.39 | 73.65 | 98.96 | 95.48 | 85.52 | 77.37 | 84.18 | 81.52 |
| Binoculars | 94.75 | 88.10 | 80.00 | 74.76 | 98.26 | 94.87 | 93.80 | 88.32 | 91.70 | 86.51 |
| Fast-DetectGPT | 79.56 | 72.45 | 77.18 | 70.13 | 84.43 | 74.45 | 65.23 | 60.53 | 76.60 | 69.39 |
| **Avg.** | 77.67 | 68.24 | 69.72 | 66.96 | 87.89 | 84.01 | 73.96 | 67.77 | 77.30 | 71.67 |
| **Supervised Detectors** | | | | | | | | | | |
| Rob-Base | 99.77 | 98.10 | 89.82 | 80.98 | 99.99 | 99.65 | 99.81 | 98.51 | 97.34 | 94.31 |
| Rob-Large | 99.77 | 98.95 | 87.01 | 80.42 | 99.99 | 99.95 | 99.95 | 99.20 | 96.68 | 94.63 |
| X-Rob-Base | 98.36 | 96.20 | 81.93 | 75.06 | 99.96 | 99.30 | 93.47 | 90.62 | 93.43 | 90.29 |
| X-Rob-Large | 99.79 | 98.31 | 89.07 | 80.32 | 99.99 | 99.90 | 99.82 | 99.20 | 97.21 | 94.43 |
| **Avg.** | 99.42 | 97.89 | 86.95 | 79.19 | 99.98 | 99.70 | 98.26 | 96.88 | 96.16 | 93.41 |

We explored a critical question in real-world detection: How do human-driven factors impact detector performance? To investigate this, we simulated various modifications to human-written texts. We introduced paraphrase attacks to mimic text revisions and incorporated spelling errors through perturbation attacks. Moreover, we mixed LLM-generated sentences with human-written content to simulate AI-assisted writing scenarios. Experimental results, as shown in Table 6, indicate that attacks on human-written texts yield markedly different outcomes compared to those on LLM-generated texts. Specifically, paraphrasing attacks on human-written texts effectively confused zero-shot detectors, reducing the AUROC by an average of 7.95%. In contrast, data mixing had a minimal impact on zero-shot detectors' performance, with only a slight decline of 3.71% in AUROC. This contrasts sharply with the significant 18.17% decline in AUROC when human-written texts were mixed with LLM-generated texts. The resilience of human-written texts to such mixing may be attributed to their inherent complexity, making it difficult for zero-shot detectors to identify the inclusion of LLM-generated content. Interestingly, perturbation attacks on human-written texts appeared to enhance the discernment capabilities of zero-shot detectors, resulting in an average increase of 10.22% in AUROC. Similar trends were observed with supervised detectors. This suggests that human-written texts may inherently contain more adversarial features [41], which are utilized by detectors for identification. Such perturbations can further emphasize these distinctions, leading to improved performance.

## 4  Conclusion

In this paper, we introduce ***DetectRL***, a novel benchmark designed to evaluate the detection capabilities of detectors against LLM-generated text. ***DetectRL*** compiles texts from human sources in high-risk and abuse-prone domains, utilizes popular and powerful LLMs, employs well-designed attack techniques, and constructs datasets encompassing a diverse range of text lengths. This benchmark aims to assess the usability of detectors in scenarios that closely resemble real-world applications. Our experimental findings reveal the primary reasons why existing detectors for LLM-generated texts struggle in practical applications. Additionally, we engage in an in-depth discussion of the potential factors influencing detector performance, offering valuable insights into current detection research. Furthermore, ***DetectRL*** provides a data curation framework to facilitate the future development of LLM-generated text detection technologies. This framework supports the rapid creation of an evolving, comprehensive, and adversarial benchmark, enabling continuous adaptation and improvement of detectors in the ongoing cat-and-mouse game of LLM-generated text detection.

## Acknowledgments

This work was supported in part by the Science and Technology Development Fund, Macau SAR (Grant No. FDCT/060/2022/AFJ, the mainland China collaboration project, National Natural Science Foundation of China Grant No. 62261160648), the Research Program of Guangdong Province (Grant No. 2220004002576, EF2023-00090-FST), the Science and Technology Development Fund, Macau SAR (Grant No. FDCT/0070/2022/AMJ, the mainland China collaboration project, China Strategic Scientific and Technological Innovation Cooperation Project Grant No. 2022YFE0204900), the Multi-year Research Grant from the University of Macau (Grant No. MYRG-GRG2023-00006-FST-UMDF, MYRG-GRG2024-00165-FST), and the Tencent AI Lab Rhino-Bird Gift Fund (Grant No. EF2023-00151-FST).

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

# A  Related work

## A.1  LLM-generated text and risk

With the expansion of model size [42] and the development of efficient preference alignment methods [43, 44], LLMs have emerged with powerful capabilities for text understanding and generation [45, 46]. The text produced by the current LLMs closely resembles the quality of human-written text, particularly in terms of coherence, fluency, and grammatical accuracy, making it difficult for humans to distinguish between the two [47]. The release of ChatGPT has propelled human society into the era of LLMs, with these models finding widespread application across various aspects of daily life, such as generating advertising copy [48], writing news articles [49], storytelling [50], and coding [51]. They are also significantly influencing various fields and industries, including education [52], law [53], and medicine [54], gaining broad acceptance among people.

However, the use of LLMs has raised several concerns. Recent research by [4] highlights significant challenges and potential risks associated with LLM-generated text from five perspectives: regulatory oversight related to artificial intelligence and copyright [55], erosion of user trust in internet content, homogenization of generated text that could impede LLM progress [56], challenges posed to education and academia by LLM misuse [57], and the formation of information echo chambers in society.

## A.2  LLM-generated text detection

Given the potential misuse of LLMs, it is crucial to develop detectors that can effectively identify LLM-generated text. These detectors can help minimize the threats posed by misuse, thereby promoting the trustworthy AI applications in the era of LLMs [58, 59]. Existing LLM-generated text detection technologies [4, 60] mainly includes watermarking technology, statistics-based methods, neural-based detectors and human-assisted methods. Despite the impressive progress in LLM-generated text detection task, [23, 61] point out that these detectors become unreliable when under real-world scenarios and well-designed attacks. Building more effective and robust detectors remains a significant challenge.

## A.3  Detection benchmark

Previous work has already dedicated significant effort to the construction of benchmarks for LLM-generated text detection, mainly encompassing early deepfake research such as the TweepFake dataset [62] and the GROVER Dataset [63], as well as prior work on detecting LLM-generated texts like the GPT-2 Output Dataset[6] and TuringBench [8]. HC3 [47] is one of the recent impressive datasets, containing ChatGPT-generated text data in both English and Chinese, covering multi-domain and multi-lingual evaluation. Other benchmarks, such as MGTBench [9], ArguGPT [64], and the MAGE [11], also consider texts generated by various LLMs. M4 [65] is a comprehensive dataset recently released, covering multi-domain, multi-lingual, and multi-generator evaluation scenarios.

However, these benchmarks still mainly focused on ideal detection settings, such as using some open-source language models with limited performance and simple text generation settings, while they lack simulations and explorations of real-world application scenarios, which has been explicitly highlighted in [4]. Our work aims to bridge this gap by offering a benchmark for detecting LLM-generated texts in a form more adapted to real-world scenarios, primarily including high-risk and abuse-prone domain, the use of more powerful and commonly employed LLMs, well-designed attack methods, varied text lengths for training and testing, and factors related to real-world human writing.

# B  Limitations

Considering the rapid innovation within the NLP community, we acknowledge that our benchmark's temporal relevance could be a potential limitation. This is due to the fast-paced development of LLMs and the emergence of new application scenarios and challenges. From the perspective of LLM development, new LLMs are being created at an astonishing rate and will continue to impact existing detectors, while our benchmark only examines the detectors' ability to discriminate against the advanced and popular LLMs currently available. Regarding application scenarios and challenges,

---

[6]https://github.com/openai/gpt-2-output-dataset

newly designed attack methods, the requirement for increasingly fine-grained detection, and the ever-expanding demands of application domains place progressively higher demands on existing detectors. Our benchmark setup only examines the current major demands and does not encompass the full spectrum of challenges, including those that may arise in the future.

Nonetheless, we open-source our benchmark framework and encourage researchers to build upon it. By using our framework, researchers can quickly create more applicable and demand-specific test data to evaluate detector performance, ensuring the benchmark remains relevant as the field evolves.

## C  Ethics statement

We developed *DetectRL* by collecting publicly available human-written texts in high-risk and abuse-pron domains, generating similar texts using advanced and popular LLMs, designing and applying various attack methods for data augmentation. The release of *DetectRL* aims to advance research on detecting LLM-generated texts, enhancing their robustness and applicability of detectors in real-world scenarios. However, while promoting this research, we have also considered the potential for misuse. By making our dataset construction framework publicly available, there's a possibility that our well-designed attack methodologies could be used to develop defenses that might undermine existing detection systems.

Despite this risk, we believe that our work will significantly contribute to the development of more robust and applicable detectors for LLM-generated text. These detectors can be continuously improved and employed to enhance LLM-generated text applications in the era of LLMs, all while participating in the ongoing cat-and-mouse game with evolving attack methods.

Additionally, although we have manually reviewed most of the data, there remains a risk that the data may still contain personally identifiable information or offensive content. Therefore, please ensure that our data is used solely for academic purposes and exercise caution.

## D  Data collection

### D.1  Human-written datasets

The human-written texts we utilized were sourced from domains where real-world applications of LLMs present higher risks. We selected Arxiv Abstracts to represent academic writing, Xsum for news writing, Writing Prompts for creative writing, and Yelp Reviews for social media interactions. The specific details of these datasets are as follows:

**ArXiv Abstracts**  The ArxivPapers dataset[7] is an unlabelled collection of over 104K papers related to machine learning published on arXiv.org between 2007 and 2020. The dataset includes around 94K papers (with available LaTeX source code) organized into a structured format comprising titles, abstracts, sections, paragraphs, and references.

**XSum**  The Extreme Summarization dataset serves as a benchmark for evaluating abstractive single-document summarization systems. This collection includes 226,711 news articles sourced from BBC reports between 2010 and 2017, covering a diverse range of topics such as news, politics, sports, weather, business, technology, science, health, family, education, entertainment, and the arts [17].

**Writing Prompts**  The Writing Prompts dataset is a dataset focused on the art of story generation, comprising 300,000 human-written stories, each paired with a unique writing prompt from an online community. This extensive collection is designed to support hierarchical story generation, a process that starts with creating a story premise and evolves into a complete narrative [18].

**Yelp Reviews**  The Yelp Reviews Polarity dataset originates from the Yelp Dataset Challenge 2015,[8] featuring reviews posted on Yelp. Refined by [19] for text classification research, the dataset categorizes reviews with 1 and 2 stars as negative (class 1) and those with 3 and 4 stars as positive

---

[7]https://www.kaggle.com/datasets/spsayakpaul/arxiv-paper-abstracts/data
[8]http://www.yelp.com/dataset_challenge

(class 2), providing a balanced approach to sentiment analysis. It includes a total of 560,000 training samples and 38,000 testing samples.

## D.2 Generative models and hyper-parameters

The generative models we selected are powerful LLMs commonly used in daily life. Table 7 lists the model paths or API services of these LLMs. The temperature for all models is set to the default parameter of 1, promoting the generation of creative and unpredictable text. The specific details of these LLMs are as follows:

Table 7: Details of the generative models that is used to produce LLM-generated text.

| Generative Model | Model Path / API Service | Hyper-parameters |
|---|---|---|
| GPT-3.5-turbo | OpenAI/gpt-3.5-turbo | temperature=1 |
| PaLM-2-bison | Google/chat-bison@002 | temperature=1 |
| Claude-instant | Anthropic/claude-instant-1.2 | temperature=1 |
| Llama-2-70b | meta-llama/Llama-2-70b-chat-hf | temperature=1 |

**GPT-3.5-turbo**   GPT-3.5-turbo [66], developed by OpenAI, is a variant of the Generative Pre-trained Transformer (GPT) model, specifically tailored for generating human-like text based on the input it receives. This model has been trained on a diverse range of internet text, enabling it to understand and produce responses across a vast array of topics and styles.

**PaLM-2-bison**   PaLM-2-bison [14] represents the latest advancement in Google's LLMs technology, building upon the foundation of PaLM [67]. This model showcases exceptional capabilities in advanced reasoning tasks such as code interpretation and mathematical problem-solving, classification and question-answering, adept translation, and multilingual communication, as well as in generating natural language with improved proficiency over previous models.

**Claude-instant**   Claude-instant [15] represents a significant leap forward in the realm of AI assistants, developed from Anthropic's rigorous research into crafting AI systems that are helpful, honest, and harmless. Designed to accommodate a wide array of use cases, Claude excels in summarization, search functionalities, creative and collaborative writing, question answering, and coding, among other tasks.

**Llama-2-70b**   Llama-2-70b is a SOTA generative open-source LLMs developed by Meta, part of the broader Llama 2 collection [42]. This model outperforms numerous open-source chat models in benchmark evaluations and equates to the leading closed-source models like ChatGPT and PaLM in terms of helpfulness and safety.

## D.3 Data generation settings

All text generation tasks were conducted through chat with LLMs. Specifically, for academic writing abstracts, we provided the article's title to the LLMs and asked them to generate an abstract based on the title; for news articles, we provided the summary of the article and asked the LLMs to generate the complete news article based on the summary; for creative writing, we provided writing prompts to the LLMs and requested that they engage in creative storytelling based on these prompts. Social media was the simplest task, as the LLMs would continue writing based on the first sentence of the social commentary text. Below, we provide the generation instructions for texts in different domains:

### D.3.1 Academic writing

```
[
    {'role': 'user', 'content': 'Given the academic article title, write
an academic article abstract with <sentences num> sentences:\n academic
article title: <prefix> \n academic article abstract:'},
]
```

1: Direct prompt for academic writing

The <sentences num> refers to the sentences length corresponding to the human-written sample, and <prefix> is the specific article title. For example, it could be like "Calculation of prompt diphoton production cross sections at Tevatron and LHC energies", and the response is supposed to write an academic abstract based on the sentences length and article title of the human-written sample.

### D.3.2 News writing

```
[
    {'role': 'user', 'content': 'Given the news article summary, write a
news article with <sentences num> sentences:\n news article summary: <
prefix> \n news article:'},
]
```

2: Direct prompt for news writing

The <sentences num> refers to the sentences length corresponding to the human-written sample, and <prefix> is the specific news article summary. For example, it could be like "A former Lincolnshire Police officer carried out a series of sex attacks on boys, a jury at Lincoln Crown Court was told.", and the response is supposed to write a news article based on the sentences length and the news article summary of the human-written sample.

### D.3.3 Creative writing

```
[
    {'role': 'user', 'content': 'Given the writing prompt, write a story
with <sentences num> sentences: \n writing prompt: <prefix> \n story:'},
]
```

3: Direct prompt for creative writing

The <sentences num> refers to the sentences length corresponding to the human-written sample, and <prefix> is the specific writing prompt. For example, it could be like "Through Iron And Flame", and the response is supposed to write a story based on the sentences length and the writing prompt of the human-written sample.

### D.3.4 Social media

```
[
    {'role': 'user', 'content': 'Given the review\'s first sentence,
please help to continue the review with <sentences num> sentences:\n
review's first sentence: <prefix> \n continued review:'},
]
```

4: Direct prompt for social media

The <sentences num> refers to the sentences length corresponding to the human-written sample, and <prefix> is the specific writing prompt. For example, it could be like "I don't know what Dr. Goldberg was like before moving to Arizona, but let me tell you, STAY AWAY from this doctor and this office.", and the response is supposed to write the continued review based on the sentences length and the first sentence of the human-written sample.

## D.4 Data attacks settings

### D.4.1 Prompt attacks

Prompt attacks are designed to use carefully crafted prompts to guide LLMs to generate text that aligns more closely with human writing styles. The Prompt Attacks we use include Few-Shot Prompt [21] and ICO Prompt (part of SICO Prompt) [22]. Few-Shot Prompting involves presenting LLMs with a few human-written examples to enhance alignment with human writing styles. The SICO Prompt introduces a novel approach called Substitution-based In-Context Example Optimization (SICO), which automatically constructs prompts to evade detection, as proposed by [22]. It operates through a

Table 8: Data attacks settings.

| Attacks Typts | Sub Types | Methods |
|---|---|---|
| Direct Prompt | Direct Prompt | Prompt |
| Prompt Attacks | Few-Shot Prompt | Prompt |
| | ICO Prompt | Prompt |
| Paraphrase Attacks | DIPPER Paraphrase | DIPPER Paraphraser |
| | Polish Using LLMs | Prompt |
| | Back Translation | Google Translation API |
| Perturbation Attacks | Character-Level Perturbation | TextFooler |
| | Word-Level Perturbation | DeepBugWord |
| | Sentence-Level Perturbation | TextBugger |
| Data Mixing | Multi-LLMs Mixing | Sentence Mixing |
| | LLM-Centered Mixing | Sentence Mixing |

two-stage prompting process. We specifically use the ICO (In-Context Example Optimization) aspect of SICO, excluding the substitution process to prevent text perturbations. We provide examples of Few-Shot Prompting and ICO Prompting for academic writing tasks as fellow:

```
[
    {'role': 'user', 'content': '<in content learning examples> \n Given
the academic aticle title, write an academic aticle with <{sentences num>
 sentences: \n academic aticle title: <prefix>: \n academic aticle:'},
]
```

5: Few-Shot Prompt

The <In-content learning examples> refer to contextual examples retrieved for LLMs to learn from. We set the number of examples to three, using the BM25 retrieval algorithm. Each example includes an academic article title and a corresponding article pair. The <sentences num> refers to the sentences length of the corresponding human-written sample, the <prefix> is the specific article title, and the task is to write an academic article abstract based on the sentence length and article title of the human-written sample.

```
[
    {'role': 'user', 'content': 'Here are the writings from AI and human:
 \n <in-content learning examples> \n Compare and give the key distinct
feature (specifically vocabulary, sentence structure) of human\'s
writings (do not use examples):'},
    {'role': 'bot', 'content': '<step1 response>'},
    {'role': 'user', 'content': 'Based on the description, given the
academic article title, write an academic article with <sentences num>
sentences in human style writings: \n academic article title: <prefix> \n
 human:'},
]
```

6: ICO Prompt

Similar to Few-Shot Prompt, the <in - content learning examples > in the ICO Prompt refer to context examples retrieved for the LLM to learn from. We set the number of examples to 3, using the BM25 retrieval algorithm. Each example consists of text generated by an LLM and text written by a human. <step1 response> refers to the answer from the first round of questioning, where the model extracts key distinct features of human writings. The <sentences num> refers to the word length of the corresponding human writing sample, and <prefix> is the specific article title.

### D.4.2 Paraphrase attacks

Paraphrase attacks involve rewriting text to preserve its original meaning. We utilize various techniques, including the DIPPER paraphrasing tool [23], Back-translation, and Polishing with LLMs. The Discourse Paraphraser (DIPPER), as described by [23], is an advanced 11-billion parameter

model designed for generating paraphrases by considering context and managing lexical diversity and content order. Inspired by real-world applications, we implement machine translation for paraphrasing through back-translation. Specifically, we use the Google Translate API[9] to translate each LLM-generated sample from English to Chinese and then back to English. To ensure that the text maintains good semantic consistency before and after Back-translation, we use BERTScore [68], an automatic evaluation metric that assesses translation quality from a semantic perspective. Polishing with LLMs is a widely used paraphrasing method in the era of LLMs, typically initiated via prompts. Below, we provide an example of polishing with LLMs for academic writing tasks:

```
[
    {'role': 'user', 'content': 'Given the academic article abstract,
polish the writing to meet the review style, improve the spelling,
grammar, clarity, concision and overall readability: \n academic article
abstract: <prefix>'},
]
```

7: Polish Prompt

The <prefix> is the specific article abstract, and the response is supposed to polish the provided academic article abstract.

Table 9: Datasets Statistics. **FP** stands for Few-Shot Prompt, **IP** stands for ICO Prompt; **DP** represents DIPPER paraphrase, **PP** represents Polishing with LLMs, **BP** represents Back-translation paraphrase; **CP** stands for Character-Level perturbation, **WP** stands for Word-Level perturbation, **SP** stands for Sentence-Level perturbation; **MM** represents Multi-LLMs Mixing, **LM** represents LLM-Centered Mixing.

| Domains &Datasets | Channel | Direct | Prompt Attacks | | Paraphrase Attacks | | | Perturbation Attacks | | | Data Mixing | | Total |
|---|---|---|---|---|---|---|---|---|---|---|---|---|---|
| | | | FP | IP | DP | PP | BP | CP | WP | SP | MM | LM | |
| Arxiv Abstracts | Human | 2,800 | - | - | 2,800 | 2,800 | 2,800 | 2,800 | 2,800 | 2,800 | - | 2,800 | 25,200 |
| | GPT-3.5-turbo | 700 | 700 | 700 | 700 | 700 | 700 | 700 | 700 | 700 | 700 | 700 | 8,400 |
| | Claude-instant | 700 | 700 | 700 | 700 | 700 | 700 | 700 | 700 | 700 | 700 | 700 | 8,400 |
| | PaLM-2-bison | 700 | 700 | 700 | 700 | 700 | 700 | 700 | 700 | 700 | 700 | 700 | 8,400 |
| | Llama-2-70b | 700 | 700 | 700 | 700 | 700 | 700 | 700 | 700 | 700 | 700 | 700 | 8,400 |
| XSum | Human | 2,800 | - | - | 2,800 | 2,800 | 2,800 | 2,800 | 2,800 | 2,800 | - | 2,800 | 25,200 |
| | GPT-3.5-turbo | 700 | 700 | 700 | 700 | 700 | 700 | 700 | 700 | 700 | 700 | 700 | 8,400 |
| | Claude-instant | 700 | 700 | 700 | 700 | 700 | 700 | 700 | 700 | 700 | 700 | 700 | 8,400 |
| | PaLM-2-bison | 700 | 700 | 700 | 700 | 700 | 700 | 700 | 700 | 700 | 700 | 700 | 8,400 |
| | Llama-2-70b | 700 | 700 | 700 | 700 | 700 | 700 | 700 | 700 | 700 | 700 | 700 | 8,400 |
| Writing Prompts | Human | 2,800 | - | - | 2,800 | 2,800 | 2,800 | 2,800 | 2,800 | 2,800 | - | 2,800 | 25,200 |
| | GPT-3.5-turbo | 700 | 700 | 700 | 700 | 700 | 700 | 700 | 700 | 700 | 700 | 700 | 8,400 |
| | Claude-instant | 700 | 700 | 700 | 700 | 700 | 700 | 700 | 700 | 700 | 700 | 700 | 8,400 |
| | PaLM-2-bison | 700 | 700 | 700 | 700 | 700 | 700 | 700 | 700 | 700 | 700 | 700 | 8,400 |
| | Llama-2-70b | 700 | 700 | 700 | 700 | 700 | 700 | 700 | 700 | 700 | 700 | 700 | 8,400 |
| Yelp Reviews | Human | 2,800 | - | - | 2,800 | 2,800 | 2,800 | 2,800 | 2,800 | 2,800 | - | 2,800 | 25,200 |
| | GPT-3.5-turbo | 700 | 700 | 700 | 700 | 700 | 700 | 700 | 700 | 700 | 700 | 700 | 8,400 |
| | Claude-instant | 700 | 700 | 700 | 700 | 700 | 700 | 700 | 700 | 700 | 700 | 700 | 8,400 |
| | PaLM-2-bison | 700 | 700 | 700 | 700 | 700 | 700 | 700 | 700 | 700 | 700 | 700 | 8,400 |
| | Llama-2-70b | 700 | 700 | 700 | 700 | 700 | 700 | 700 | 700 | 700 | 700 | 700 | 8,400 |
| Totall | - | 22,400 | 11,200 | 11,200 | 22,400 | 22,400 | 22,400 | 22,400 | 22,400 | 22,400 | 11,200 | 22,400 | **235,200** |

### D.4.3   Perturbation attacks

Perturbation attacks primarily focus on adversarial perturbations on the text directly generated by LLMs, effectively simulating post-processing of LLM-generated text by humans and common writing errorslike spelling mistakes in real life. Our approach employs adversarial perturbation methods include TextFooler [25], DeepWordBug [24], and TextBugger [24], which correspond to word-level, character-level, and sentence-level adversarial perturbations, respectively. DeepWordBug [24] is a black-box perturbation method that can efficiently generate character-level text perturbations with the goal of minimizing the edit distance of the perturbation. TextFooler [25] is a text-based adversarial method that uses synonyms to replace words in a sentence that are vulnerable to attacks while maintaining good grammatical correctness and semantic coherence. TextBugger [24] creates adversarial texts suitable for real-world applications, ensuring that the adversarial samples remain visually and semantically consistent with the originals, and considers both character and word-level perturbations. All perturbation attacks are implemented using the TextAttacks [26] framework.

---

[9]https://translate.google.com/

### D.4.4 Data mixing

Data mixing is a common real-world scenario. Our data mixing methods include a mixing of texts generated by various LLMs (Multi-LLMs Mixing) and texts centered around LLM-generated text with a mixing of human-written text (LLM-Centered Mixing). The Multi-LLMs Mixing refers to a single text composed of sentences from different generative models. LLM-Centered Mixing involve replacing one quarter of the sentences in an LLM-generated text with human-written text at random. To facilitate this, we ensured that both human-written and LLM-generated texts contained at least four sentences during collection, providing a solid foundation for our data mixing process.

For Multi-LLMs Mixing, we sample and recombine sentences from texts generated by four different LLMs, aligning with the length of human-written texts to create a new sample. Similarly, for LLM-Centered Mixing, one-quarter of the sentences in the LLM-generated text are randomly replaced with sentences from the corresponding human-written text. This approach presents a more challenging scenario, as the data-mixed samples often lack coherent semantics.

### D.5 Datasets statistics

The statistics for the curated datasets are presented in Table 9. The datasets include 100,800 human-written samples, consisting of 11,200 raw samples and 89,600 that have undergone attack manipulations. Additionally, there are 134,400 samples generated by LLMs, categorized as follows: 11,200 with direct prompt, 22,400 with prompt attacks, 33,600 with paraphrase attacks, 33,600 with perturbation attacks, and 22,400 involving data mixing.

### D.6 Textual features analysis

In this section, we analyze the textual features of ***DetectRL*** samples to provide additional potentially valuable insights.

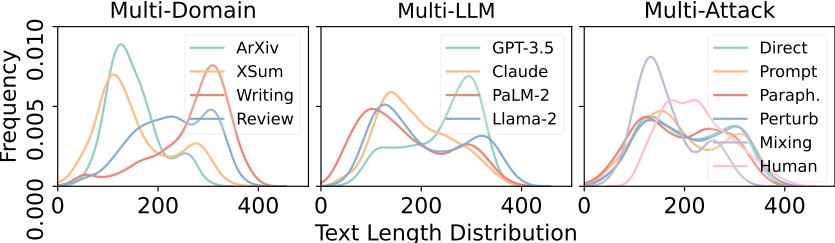

Figure 4: Text length distribution of ***DetectRL***.

**Text length** We performed a statistical analysis of text length distribution in ***DetectRL***, as shown in Figure 4. Compared to academic writing and social media texts, news writing and creative writing exhibit notably longer average lengths. The distributions for texts generated by Claude-instant, PaLM-2-bison, and Llama-2-70b are similar, whereas GPT-3.5-turbo tends to produce longer texts. Additionally, we observed that samples subjected to attack manipulation show almost no significant difference in length, except in the data mixing setup.

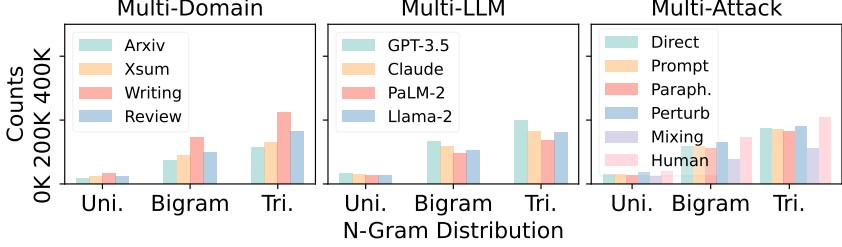

Figure 5: *N*-gram distribution of ***DetectRL***.

**N-grams** We performed statistical analysis of the *n*-gram distribution in **DetectRL**, focusing on unigrams, bigrams, and trigrams. The results are presented in Figure 5. Among the four domains, creative writing exhibits the greatest variety of unigrams, bigrams, and trigrams, indicating a higher *n*-gram diversity. In contrast, academic writing shows the lowest diversity. Among the different LLMs, GPT-3.5-turbo demonstrates the most extensive vocabulary usage, followed by Claude-instant, Llama-2-70b, and PaLM-2-bison, in order of decreasing *n*-gram richness. Additionally, samples with perturbation attack show the highest *n*-gram diversity due to the substitution of characters and vocabulary. Notably, in samples involving data mixing, *n*-gram richness is significantly lower, approximately half that of other sample types.

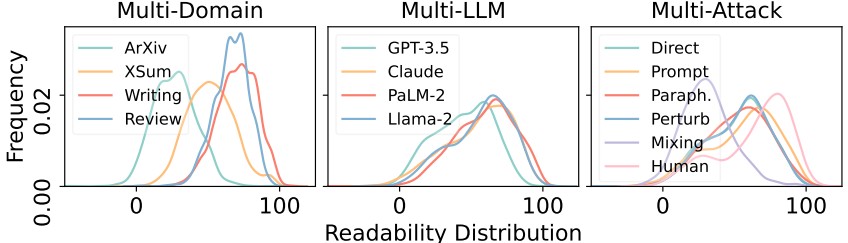

Figure 6: Readability distribution of **DetectRL**.

**Readability** We carried out a statistical analysis of the readability distribution within **DetectRL**, primarily by calculating the Flesch Reading Ease Score (FRES) for each sample. The FRES [69] assesses reading difficulty by considering word length and sentence length. The formula used to calculate this score is as follows:

$$
\begin{aligned}
\text{FRES} = 206.835 - 1.015 \times &\left( \frac{\text{Total Words}}{\text{Total Sentences}} \right) \\
&- 84.6 \times \left( \frac{\text{Total Syllables}}{\text{Total Words}} \right)
\end{aligned}
\tag{1}
$$

Scores range from 0 to 100, with higher scores indicating better readability. The results revealed significant differences in text readability across various domains. Among all categories, creative writing texts exhibit the highest readability, followed by social media and news writing texts, while academic writing texts are the least readable. When comparing texts generated by different LLMs, we observed that the texts produced by Claude-instant, PaLM-2-bison, and Llama-2-70b show a high degree of consistency in readability. However, texts generated by GPT-3.5-turbo show a noticeable readability gap compared to the others. Similarly, texts generated from direct prompts, prompt attacks, paraphrase attacks, and perturbation attacks display a comparable distribution of readability scores. Yet, samples processed through data mixing show lower average readability, likely due to the inclusion of human-written texts.

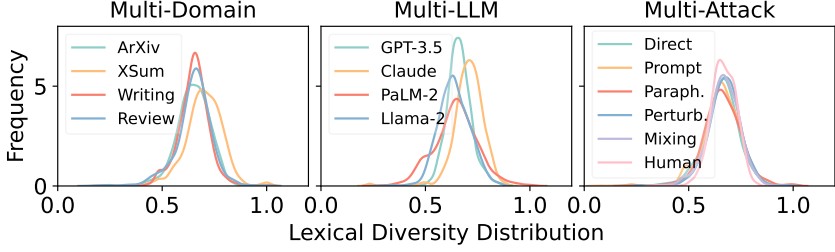

Figure 7: Lexical diversity distribution of **DetectRL**.

**Lexical diversity** We conducted a statistical analysis of the Lexical Diversity Score (LDS) within **DetectRL**, using the following formula to calculate this feature:

$$LDS = \frac{\text{Number of Unique Word Types}}{\text{Total Number of Words}} \qquad (2)$$

Upon examining different dimensions within **DetectRL**, we observed a unique phenomenon in Figure 7: there is almost no significant variation in the distribution of lexical diversity across domains and attacks. This contrasts sharply with the other three features we analyzed. Additionally, we noted that although GPT-3.5-turbo generated samples exhibited slight differences compared to those samples generated by various other LLMs, they generally clustered around a lexical diversity score of 0.7.

# E  Experiment settings

## E.1  Black-box detection task settings

The focus of our research is to develop a detector assessment benchmark that closely aligns with real-world usage scenarios. A crucial prerequisite is black-box detection, as texts encountered in reality often originate from unknown sources. Therefore, we designed all our experiments as black-box text detection tasks. In this context, all detectors assessed does not have access to the source model used for text generation.

The setting of the black-box detection task significantly distinguishes our experiments from traditional LLM-generated text detection experiments. In this paper, we follow the experimental settings of [32] and use GPT-Neo-2.7B [38] as a surrogate model for the traditional zero-shot method. For the supervised method, we train the detector using data specific to each task setting. In the generalization experiment, we assess the detector's ability to recognize and handle text generated by different domains and models, testing its performance on data from other domains and generative models.

## E.2  Detectors settings

Table 10: Performance of prompt attacks and data mixing.

| Settings → Detectors ↓ | Few-Shot Prompt | | | | ICO Prompt | | | | Muti-models | | | | LLM-centered | | | |
|---|---|---|---|---|---|---|---|---|---|---|---|---|---|---|---|---|
| | Pre | Rec | $F_1$ | AUROC | Pre | Rec | $F_1$ | AUROC | Pre | Rec | $F_1$ | AUROC | Pre | Rec | $F_1$ | AUROC |
| **Zero-shot Detectors** | | | | | | | | | | | | | | | | |
| Log-Likelihood | 90.60 | 71.72 | 80.06 | 89.50 | 89.09 | 65.67 | 75.61 | 85.10 | 86.57 | 77.40 | 81.73 | 89.94 | 74.83 | 56.64 | 64.48 | 74.92 |
| Entropy | 00.00 | 00.00 | 00.00 | 23.38 | 50.02 | 100.0 | 66.68 | 28.88 | 00.00 | 00.00 | 00.00 | 44.13 | 50.02 | 100.0 | 66.68 | 37.36 |
| Rank | 82.50 | 69.24 | 75.29 | 84.16 | 78.65 | 65.07 | 71.22 | 80.17 | 75.84 | 78.20 | 77.00 | 84.05 | 60.78 | 50.89 | 55.39 | 61.06 |
| Log-Rank | 84.59 | 77.87 | 81.09 | 89.10 | 91.45 | 63.69 | 75.08 | 84.65 | 85.32 | 79.10 | 82.09 | 82.81 | 74.38 | 57.34 | 64.76 | 74.62 |
| LRR | 83.43 | 70.93 | 76.67 | 82.74 | 91.10 | 57.93 | 70.83 | 80.31 | 85.22 | 67.50 | 75.33 | 85.19 | 79.27 | 45.53 | 57.84 | 71.33 |
| NPR | 76.55 | 71.00 | 73.67 | 79.76 | 74.50 | 63.85 | 68.77 | 76.41 | 77.00 | 67.50 | 71.94 | 79.46 | 58.90 | 42.82 | 49.59 | 55.60 |
| DetectGPT | 64.94 | 25.02 | 36.12 | 48.16 | 81.38 | 30.38 | 44.25 | 56.65 | 69.95 | 29.12 | 41.13 | 52.97 | 61.83 | 12.73 | 21.12 | 46.32 |
| DNA-GPT | 81.03 | 77.97 | 79.47 | 87.50 | 79.02 | 72.12 | 75.41 | 84.66 | 78.13 | 84.00 | 80.96 | 87.93 | 62.48 | 65.27 | 63.85 | 66.30 |
| Revise-Detect. | 82.48 | 83.13 | 82.80 | 90.13 | 80.61 | 67.65 | 73.57 | 81.38 | 78.26 | 81.00 | 79.60 | 87.40 | 72.71 | 63.19 | 67.62 | 76.35 |
| Binoculars | 96.44 | 75.29 | 84.56 | 92.09 | 96.92 | 75.00 | 84.56 | 88.88 | 95.20 | 87.40 | 91.13 | 95.57 | 87.48 | 73.51 | 79.89 | 86.15 |
| Fast-DetectGPT | 82.35 | 47.22 | 60.02 | 68.87 | 84.11 | 53.57 | 65.45 | 73.28 | 82.21 | 60.10 | 69.43 | 77.14 | 74.74 | 29.36 | 42.16 | 59.30 |
| Avg. | 74.99 | 60.85 | 66.34 | 75.94 | 81.53 | 64.99 | 70.13 | 74.57 | 73.97 | 64.66 | 68.21 | 78.78 | 68.85 | 54.29 | 57.58 | 64.48 |
| **Supervised Detectors** | | | | | | | | | | | | | | | | |
| Rob-Base | 98.29 | 97.02 | 97.65 | 99.38 | 97.45 | 98.71 | 98.07 | 99.75 | 98.00 | 98.40 | 98.20 | 99.81 | 97.07 | 95.53 | 96.30 | 96.32 |
| Rob-Large | 99.00 | 98.51 | 98.75 | 99.88 | 99.20 | 99.20 | 99.20 | 99.93 | 99.00 | 99.90 | 99.45 | 99.45 | 96.04 | 98.80 | 97.40 | 97.37 |
| X-Rob-Base | 94.00 | 94.84 | 94.41 | 98.23 | 96.46 | 97.42 | 96.93 | 96.92 | 90.96 | 96.60 | 93.69 | 97.56 | 90.98 | 90.07 | 90.52 | 96.01 |
| X-Rob-Large | 98.49 | 97.51 | 98.00 | 98.01 | 98.03 | 98.80 | 98.41 | 98.41 | 99.59 | 98.90 | 99.24 | 99.25 | 97.54 | 98.71 | 98.12 | 99.64 |
| Avg. | 97.45 | 96.97 | 97.20 | 98.88 | 97.78 | 98.53 | 98.15 | 98.75 | 96.89 | 98.45 | 97.65 | 99.02 | 95.41 | 95.78 | 95.58 | 97.33 |

In this section, we introduce the different detector setups we used. For zero-shot detectors, classification thresholds are statistically derived by accessing the logits and their variants from the white-box model, utilizing the GPT-Neo-2.7B [38] to align with Fast-DetectGPT [32] experiments. Perturbation-based zero-shot methods, such as NPR [31] and DetectGPT [27], use T5-small [70] for sample perturbation. Additionally, Fast-DetectGPT employs GPT-J-6B [71] as the reference model, following the optimal settings reported by [32]. For black-box zero-shot detection methods like Revise-Detect [33] and DNA-GPT [34], we use GPT-4o-Mini [39] for operations such as text revision and continuation. For training supervised detectors, we use the parameters detailed in Table 12.

All supervised detectors were trained on a single NVIDIA GeForce RTX 3090 24GB, and all zero-shot detectors were run on a single NVIDIA A100 80GB.

Table 11: Performance of paraphrase attacks.

| Settings → Detectors ↓ | DIPPER Paraphrase | | | | Polish using LLMs | | | | Back Translation | | | |
|---|---|---|---|---|---|---|---|---|---|---|---|---|
| | Pre | Rec | $F_1$ | AUROC | Pre | Rec | $F_1$ | AUROC | Pre | Rec | $F_1$ | AUROC |
| **Zero-shot Detectors** | | | | | | | | | | | | |
| **Log-Likelihood** | 85.60 | 80.85 | 83.16 | 91.30 | 77.73 | 64.78 | 70.67 | 78.61 | 100.0 | 00.29 | 00.59 | 22.03 |
| **Entropy** | 00.00 | 00.00 | 00.00 | 32.42 | 50.02 | 100.0 | 66.68 | 32.66 | 74.36 | 72.81 | 73.58 | 79.77 |
| **Rank** | 77.91 | 73.51 | 75.65 | 82.91 | 69.43 | 63.09 | 66.11 | 72.38 | 100.0 | 00.29 | 00.59 | 24.77 |
| **Log-Rank** | 85.18 | 82.14 | 83.63 | 91.47 | 77.91 | 62.30 | 69.23 | 77.40 | 85.71 | 00.59 | 01.18 | 23.52 |
| **LRR** | 86.21 | 77.57 | 81.67 | 88.94 | 76.98 | 51.09 | 61.41 | 70.33 | 62.96 | 05.05 | 09.36 | 31.15 |
| **NPR** | 75.54 | 65.87 | 70.37 | 77.20 | 66.70 | 59.02 | 62.63 | 68.37 | 75.00 | 00.29 | 00.59 | 24.47 |
| **DetectGPT** | 66.23 | 20.70 | 31.54 | 45.90 | 72.72 | 19.84 | 31.17 | 47.83 | 00.00 | 00.00 | 00.00 | 03.81 |
| **DNA-GPT** | 82.78 | 76.78 | 79.67 | 88.33 | 73.26 | 66.07 | 69.48 | 77.33 | 100.0 | 01.68 | 03.31 | 28.77 |
| **Revise-Detect.** | 85.68 | 89.08 | 87.35 | 94.03 | 81.11 | 69.44 | 74.82 | 84.17 | 00.00 | 00.00 | 00.00 | 20.41 |
| **Binoculars** | 95.68 | 88.09 | 91.73 | 96.58 | 95.68 | 70.43 | 81.14 | 85.04 | 65.58 | 40.27 | 49.90 | 58.22 |
| **Fast-DetectGPT** | 72.92 | 63.59 | 67.93 | 75.06 | 78.42 | 37.50 | 50.73 | 61.68 | 00.00 | 00.00 | 00.00 | 17.77 |
| **Avg.** | 73.97 | 65.28 | 68.42 | 78.55 | 74.54 | 60.32 | 64.00 | 68.70 | 60.32 | 10.99 | 12.64 | 30.42 |
| **Supervised Detectors** | | | | | | | | | | | | |
| **Rob-Base** | 99.00 | 99.00 | 99.00 | 99.90 | 99.30 | 98.90 | 99.10 | 99.95 | 100.0 | 99.40 | 99.70 | 99.97 |
| **Rob-Large** | 99.70 | 99.40 | 99.55 | 99.91 | 98.62 | 99.50 | 99.06 | 99.89 | 99.90 | 99.80 | 99.85 | 99.99 |
| **X-Rob-Base** | 97.34 | 98.31 | 97.82 | 99.56 | 94.15 | 97.51 | 95.80 | 98.69 | 97.24 | 97.91 | 97.57 | 99.13 |
| **X-Rob-Large** | 98.52 | 99.60 | 99.06 | 99.77 | 98.33 | 99.30 | 98.81 | 99.93 | 100.0 | 99.50 | 99.75 | 99.75 |
| **Avg.** | 98.64 | 99.08 | 98.86 | 99.78 | 97.60 | 98.80 | 98.19 | 99.61 | 99.28 | 99.15 | 99.22 | 99.71 |

Table 12: Parameters for supervised detectors training.

| Parameters | Settings |
|---|---|
| Learning Rate | 1e-6 |
| Batch Size | 8 |
| Epochs | 3 |
| Seed | 2023 |
| GPU Envs | NVIDIA GeForce RTX 3090 24GB |

We use Youden's J statistic to determine the optimal threshold for the detectors. This approach achieves the best balance between the TPR and the FPR, thereby maximizing the overall correct classification rate.

**Log-Likelihood [30]**  A simple zero-shot method employs a language model to calculate the log-probability for each token within a text. A higher average log-likelihood indicates a greater likelihood that the text is generated by an LLM.

**Entropy [29]**  A zero-shot method relies on entropy to assess the randomness of text in order to identify text generated by LLMs. Human-written text typically shows more unpredictable variations. Consequently, text with lower entropy is more likely to have been produced by an LLM.

**Rank [30]**  A zero-shot method assigns a rank score to each token based on the previous context. By calculating the average score, a higher average rank score suggests a greater likelihood that the text is generated by an LLM.

**Log-Rank [30]**  An enhanced version of the Rank-based method. It uses a language model to calculate the logarithmic rank score of each word in the text. By calculating the average score, a higher average log-rank score suggests a greater likelihood that the text is generated by an LLM.

**LRR [31]**  The Log-Likelihood Log-Rank Ratio (LRR), an enhanced zero-shot method that effectively integrates Log-Likelihood and Log-Rank. Text with a higher LRR is more likely to be generated by an LLM.

**NPR [31]**  The Normalized Perturbed Log-Rank (NPR) is a zero-shot method that identifies differences by comparing the Log-Rank scores of perturbed human-written text with those generated by LLMs. Text with a higher NPR is more likely to be generated by an LLM.

**DetectGPT [27]** A zero-shot method for detection using probabilistic curvature. It utilizes random perturbations of paragraphs from a general pre-trained language model and discriminates LLM-generated text through the statistical curvature threshold of log probabilities.

**DNA-GPT [34]** A zeo-shot detection method that utilizes $N$-gram analysis or probability divergence in a white-box setting to compare the differences between the truncated original text and the text completed by a language model. A higher score suggests a greater likelihood that the text was generated by an LLM.

**Revise-Detect. [33]** A zero-shot black-box method based on the intuition that ChatGPT makes fewer edits to text generated by LLMs compared to human-written text. If the similarity between the text and its ChatGPT-revised version is higher, the text is more likely to be generated by an LLM.

**Binoculars [35]** A zero-shot detection method employs a pair of LLMs to calculate the ratio of perplexity to cross-perplexity. This evaluates how one model reacts to the next token predictions of another model. A lower score suggests that the text is more likely generated by an LLM.

**Fast-DetectGPT [32]** An optimized zero-shot detector that replaces the perturbation step of DetectGPT with a more efficient sampling step. We chose the optimal settings reported by the authors, using GPT-Neo-2.7b as the scoring model and GPT-J-6b [71] as the reference model.

**RoBERTa Classifier [36]** A popular and competitive detector method. Recognize LLM generated text by fine-tuning the RoBERTa classifier on large amounts of labeled text.

**XLM-RoBERTa Classifier [37]** A multi-lingual version of RoBERTa. We use XLM-RoBERTa-Base and XLM-RoBERTa-Large to build detectors to explore the potential of multilingual supervised methods.

## F    Additional experiment results

Table 13: Performance of perturbation attacks.

| Settings → Detectors ↓ | Char-Level Perturbation | | | | Word-Level Perturbation | | | | Sentence-Level Perturbation | | | |
|---|---|---|---|---|---|---|---|---|---|---|---|---|
| | Pre | Rec | $F_1$ | AUROC | Pre | Rec | $F_1$ | AUROC | Pre | Rec | $F_1$ | AUROC |
| **Zero-shot Detectors** | | | | | | | | | | | | |
| Log-Likelihood | 60.00 | 00.29 | 00.59 | 28.33 | 80.00 | 00.79 | 01.57 | 39.35 | 75.00 | 00.59 | 01.18 | 39.25 |
| Entropy | 73.65 | 70.73 | 72.16 | 77.12 | 60.30 | 69.94 | 64.76 | 62.70 | 61.51 | 72.61 | 66.60 | 65.28 |
| Rank | 00.00 | 00.00 | 00.00 | 09.68 | 00.00 | 00.00 | 00.00 | 04.38 | 00.00 | 00.00 | 00.00 | 10.42 |
| Log-Rank | 66.66 | 00.59 | 01.17 | 28.96 | 78.57 | 01.09 | 02.15 | 42.79 | 72.72 | 00.79 | 01.57 | 41.49 |
| LRR | 71.42 | 01.98 | 03.86 | 33.43 | 66.37 | 29.96 | 41.28 | 53.79 | 72.59 | 14.98 | 24.83 | 49.40 |
| NPR | 00.00 | 00.00 | 00.00 | 08.82 | 00.00 | 00.00 | 00.00 | 02.92 | 00.00 | 00.00 | 00.00 | 08.40 |
| DetectGPT | 00.00 | 00.00 | 00.00 | 16.83 | 00.00 | 00.00 | 00.00 | 16.27 | 00.00 | 00.00 | 00.00 | 22.90 |
| DNA-GPT | 88.88 | 01.58 | 03.11 | 35.44 | 94.44 | 01.68 | 03.31 | 41.35 | 81.81 | 01.78 | 03.49 | 45.54 |
| Revise-Detect. | 00.00 | 00.00 | 00.00 | 41.81 | 64.27 | 64.78 | 64.52 | 67.82 | 00.00 | 00.00 | 00.00 | 34.51 |
| Binoculars | 83.98 | 61.40 | 70.94 | 75.65 | 82.99 | 57.14 | 67.68 | 73.10 | 88.30 | 62.20 | 72.99 | 79.56 |
| Fast-DetectGPT | 73.11 | 42.36 | 53.64 | 61.99 | 63.63 | 00.69 | 01.37 | 31.69 | 73.19 | 46.32 | 56.74 | 66.49 |
| **Avg.** | 47.06 | 16.26 | 18.67 | 38.00 | 53.68 | 20.55 | 22.42 | 39.65 | 47.73 | 18.11 | 20.67 | 42.11 |
| **Supervised Detectors** | | | | | | | | | | | | |
| Rob-Base | 99.70 | 99.80 | 99.75 | 99.99 | 98.31 | 98.51 | 98.41 | 99.79 | 100.0 | 99.30 | 99.65 | 99.99 |
| Rob-Large | 99.90 | 99.80 | 99.85 | 99.96 | 99.50 | 99.90 | 99.70 | 99.98 | 100.0 | 99.40 | 99.70 | 99.97 |
| X-Rob-Base | 99.90 | 99.50 | 99.70 | 99.97 | 97.98 | 96.52 | 97.25 | 97.27 | 99.89 | 98.51 | 99.20 | 99.92 |
| X-Rob-Large | 99.90 | 99.80 | 99.85 | 99.99 | 99.50 | 99.90 | 99.70 | 99.99 | 99.89 | 99.00 | 99.45 | 99.69 |
| **Avg.** | 99.85 | 99.73 | 99.79 | 99.98 | 98.82 | 98.71 | 98.77 | 99.26 | 99.94 | 99.05 | 99.50 | 99.89 |

### F.1    Detailed robustness analysis against different types of attacks.

In this section, we will further discuss the performance of detectors against various specific attack methods. Our study on prompt attack, including Few-Shot Prompt and ICO Prompt, revealed that these methods have minimal impact on detector performance. Both zero-shot and supervised detectors showed only a 1-2% decrease in average AUROC performance. This indicates that efforts to guide models to mimic human writing through such instructions may not effectively evade detection.

For paraphrase attacks, the results in Table 11 demonstrate that DIPPER Paraphrase is not an effective tool for assessing the performance of detectors in complex black-box scenarios, as detectors nearly maintain the same identification performance as with direct prompts. However, Polishing using LLMs proves relatively effective, reducing the detector's AUROC and $F_1$ Score by approximately 10%, indicating that self-editing texts with LLMs can still diminish detection capabilities. Notably, Back-translation, a widely used paraphrasing method in our daily life, exhibits strong attack capability, significantly reducing the detector's AUROC and $F_1$ Score to 30.42% and 12.64%, respectively, a decline of over 40%.

Regarding perturbation attacks, the results in Table 13 underscore their more threatening nature compared to paraphrase attacks. On average, Character-Level, Word-Level, and Sentence-Level Perturbations resulted in a 37.75% AUROC performance decrease for detectors. Among all the attack methods we evaluated, Character-Level Perturbation ranks second only to Back-translation. Interestingly, these three types of perturbations not only showed high consistency but also revealed their potential threat in assessing detector stability. Nonetheless, all supervised detectors still performed well on datasets with the same distribution.

Lastly, we assessed detector performance in data mixing scenarios, as shown in Table 10. The results indicate that mixing texts generated by different LLMs is ineffective at confusing detectors. However, LLM-Centered Mixing poses a greater threat, particularly when a quarter of the replacement sample sentences are human-written. This leads to a performance decrease of 13.19% AUROC for the detectors.

## F.2 Details of detectors generalization performance

We conducted a comprehensive analysis of the generalization of detectors, providing detailed experimental results from three perspectives: Generalization in multi-LLM, Generalization in multi-domain, and Generalization in multi-attack. To visually present these capabilities, we have utilized heatmaps for illustration. The specific heatmaps can be found in Figure 8, Figure 9 and Figure 10.

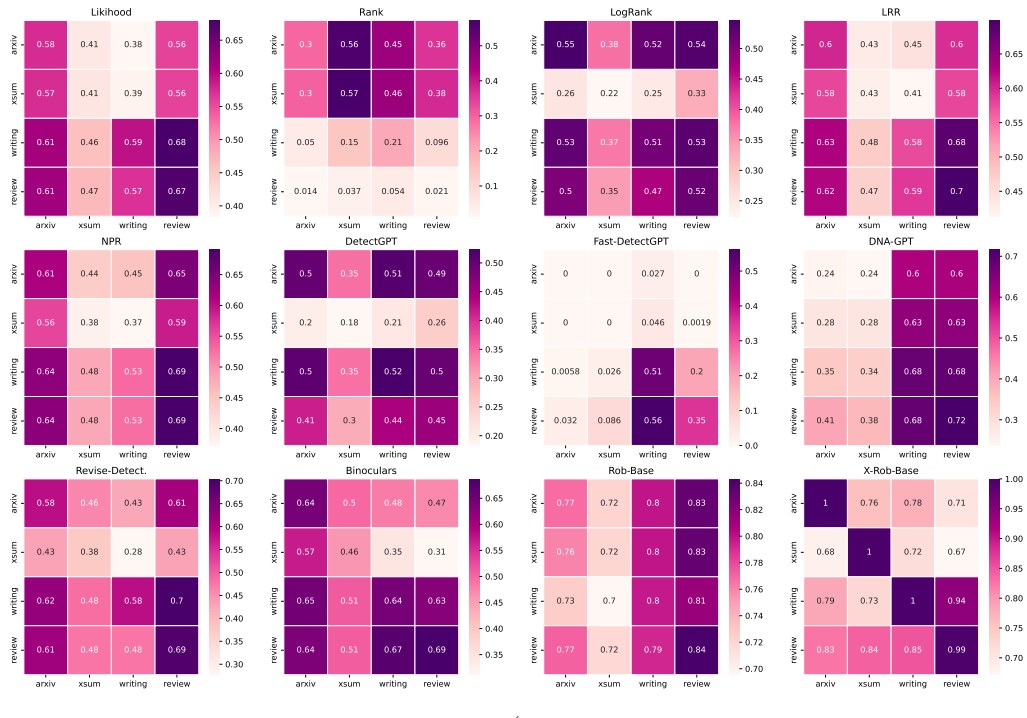

Figure 8: Generalization in multi-domain.

## F.3 Benchmark examples

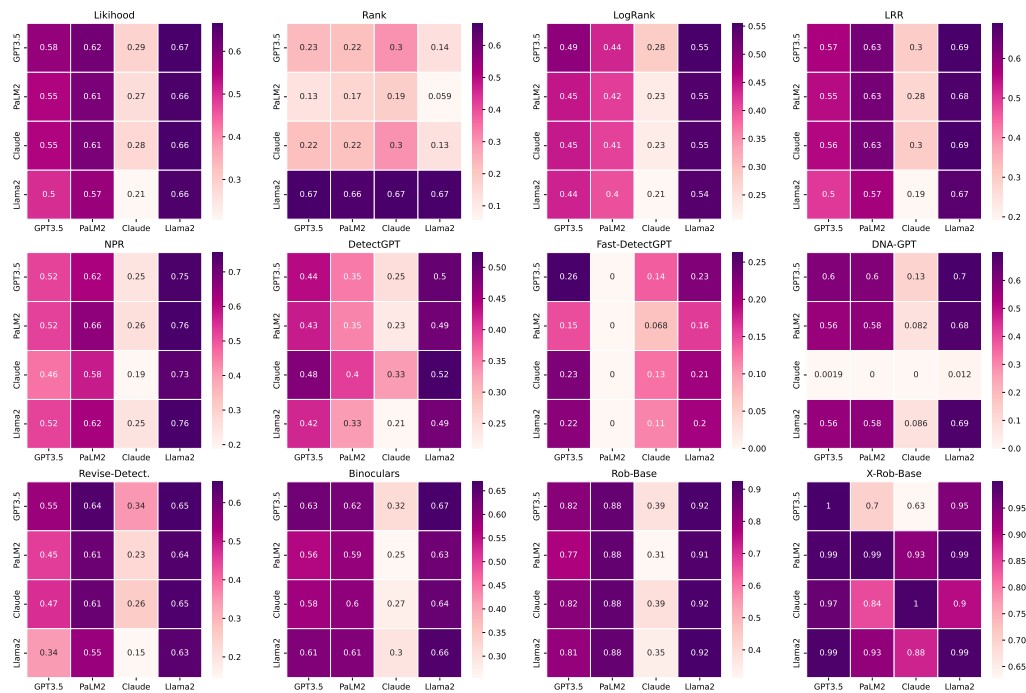

Figure 9: Generalization in multi-LLM.

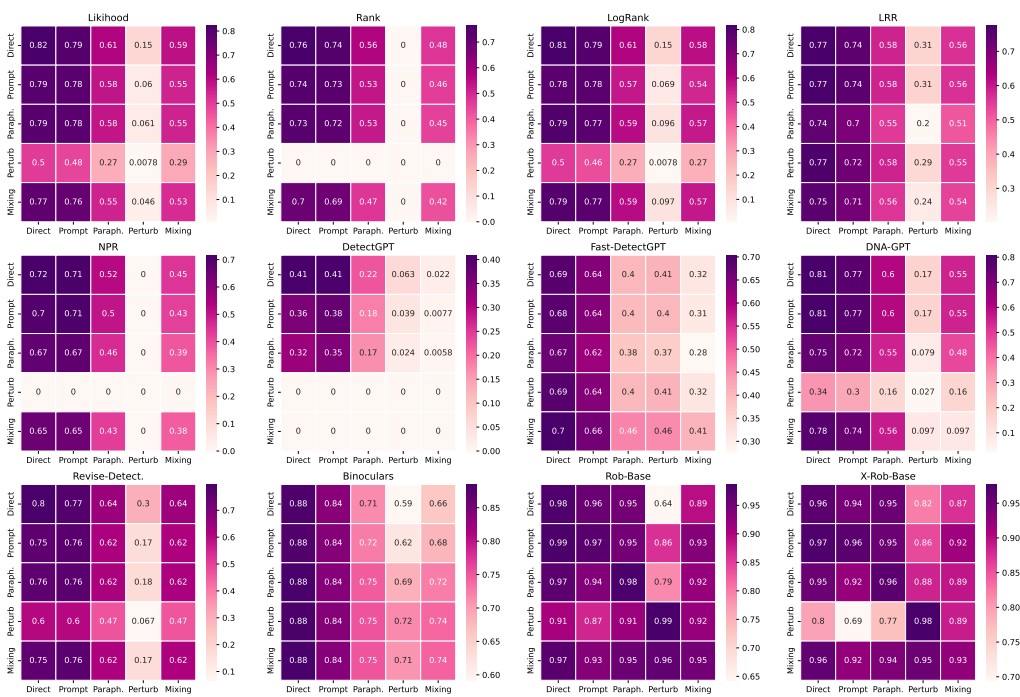

Figure 10: Generalization in multi-attack.

Table 14: Academic writing samples written by **human** in *DetectRL*. We use blue to mark human-written text, green to mark the modified parts in human-written text after paraphrase attacks, and orange to mark the modified parts in human-written text after perturbation attacks.

| Setting | Text |
| --- | --- |
| **Human-Written Text** | |
| Human | The standard C*-algebraic version of the algebra of canonical commutation relations, the Weyl algebra, frequently causes difficulties in applications since it neither admits the formulation of physically interesting dynamic laws nor does it incorporate pertinent physical observables such as (bounded functions of) the Hamiltonian. . . . |
| Polish Using LLM | The standard C*-algebraic version of the Weyl algebra, which describes canonical commutation relations, often poses challenges in applications. It hinders the formulation of physically meaningful dynamical laws and the incorporation of relevant physical observables, such as bounded functions of the Hamiltonian. . . . |
| Back Translation | Standard C*algebra version of the specification exchange algebra, Weldid numbers often cause difficulties in applications, because it neither recognizes interesting dynamic expressions, nor relevant physical observable objects, such as Hamilton, Hamilton (Limited). . . . |
| DIPPER Paraphrase | Here we present a new C*-algebra of the canonical commutation relations which does not suffer from these problems. It is based on the resolvents of the canonical operators and their algebraic relations. The resulting C*-algebra, the resolvent algebra, is shown to have many desirable analytic properties and the regularity structure of its representations is surprisingly simple. . . . |
| Character Perturbation | The standard C*-algebraec ersion of the algebra of canonical commutation relations, the Weyl algebra, frequently causes difficulties in applications since it neither admits the formulation of physically interesting dyanmicallaws nor does it incorporate pertinent physical obervables such as (bounded functions of) the Hamiltonian. . . . |
| Word Perturbation | The standard C*-algebraic version of the algebra of canonical commutation relations, the Weyl math, frequently causes hardship in applications since it neither admits the formulation of physically outstanding dynamicallaws nor does it incorporate thereto physical observables such as (bounded functions of) the Hamiltonian. . . . |
| Sentence Perturbation | The standard C*-algebraic version of the algebra of canonical commutation relations, the Weyl algera, frequently causes difficulties in applications since it neither admits the devising of physically interesting dynamical laws nor does it incorporate pertinent physical observables such as (bounded functions of) the Hamiltonian. . . . |

Table 15: Academic writing samples generated by **PaLM-2-bison** in *DetectRL*. We use pink to mark LLM-generated text, red to mark text generated by the selected prompt, green to mark the modified parts in LLM-generated text after paraphrase attacks, and orange to mark the modified parts in LLM-generated text after perturbation attacks.

| Setting | Text |
|---------|------|
| **LLM-Generated Text** | |
| Direct Prompt | The resolvent algebra is a new mathematical structure that provides a powerful framework for studying quantum systems. It is based on the idea of using the resolvent operator, which is the inverse of the energy operator, as the basic building block for constructing the algebra. . . . |
| Few-Shot Prompt | The resolvent algebra is a new approach to canonical quantum systems. It is a *-algebra of operators that is generated by the resolvent of the Hamiltonian and the identity operator. The resolvent algebra is a natural setting for the study of quantum systems, as it provides a unified framework for describing both the classical and quantum aspects of a system. . . . |
| ICO Prompt | In this paper, we introduce a new algebraic structure, called the resolvent algebra, which provides a unified framework for studying a wide class of canonical quantum systems. The resolvent algebra is a *-algebra with involution, which is generated by the resolvents of the system's Hamiltonian. . . . |
| Polish Using LLM | The resolvent algebra is a novel mathematical structure offering a robust framework for studying quantum systems. This framework is rooted in the concept of using the resolvent operator, the inverse of the energy operator, as the fundamental building block for constructing the algebra. . . . |
| Back Translation | Analysis algebra is a new mathematical structure that provides a powerful framework for studying the quantum system. It is based on the use of decomposition operators (the countdowner counts) as the basic construction block of the construction algebra. . . . |
| DIPPER Paraphrase | The resolvent algebra is a *-algebra, which means that it has a natural notion of multiplication and involution. It also has a natural topology, which makes it possible to study the structure of the algebra in a rigorous way. It is based on the idea of using the resolvent operator, which is the inverse of the energy operator, as the basic building block for constructing the algebra. . . . |
| Character Perturbation | The reolvent algebra is a new mathematical structure that provides a powerful framework for studying quantum systems. It is based on the idea of using the resolvent operator, which is the inversG of the energy operator, as the basic buildig block for constructing the algebra. . . . |
| Word Perturbation | The resolvent algebra is a new mathematical structure that provides a powerful framework for investigation quantum systems. He is based on the thought of using the resolvent operator, which is the inverse of the energy operator, as the basic building block for constructing the algebra. . . . |
| Sentence Perturbation | The resolvent algebra is a new mathemtical structure that provides a powerful framework for studying quantum systems. It is based on the idea of using the resolvent operator, which is the inverse of the energy operandi, as the basic building block for constructing the algebra. . . . |

