# OpenReview forum: "DetectRL: Benchmarking LLM-Generated Text Detection in Real-World Scenarios"
_NeurIPS.cc/2024/Datasets_and_Benchmarks_Track — NeurIPS 2024 Track Datasets and Benchmarks Poster_

### Official Review · Reviewer_tJNz · 2024-07-21
**Early work in benchmarking LLM text detectors' robustness**

**Rating:** 6
**Confidence:** 4
**Correctness:** 1. The real-world scenario claim is n…

**Review:**

Different from many other research fields related to LLMs that have a large number of benchmarks, LLM-generated text detection seems to have only a limited number of evaluation datasets. From this point of view, this paper may contribute significantly to this direction. However, I have several concerns about this paper in terms of its topic, rigorousness, and presentation.

My major concern is the main claim of this paper, which states that DetectEval is a benchmark designed for real-world scenarios. However, from my impression, the primary experiments involve detectors' performance under adversarial attacks and different input lengths. I would say it is rather controversial to conclude that the adversarial attack is a real-world use case. If the attack is sophisticated (as claimed in the paper several times), then it should seldom arise in the real world. In robustness research (particularly in CV domains), it is often believed that natural corruption is more frequent than adversarial corruption. Under the LLM-generated content scenario, I think it is more common to detect text produced by different prompts (instructions) instead of testing detectors by adding adversarial noises in a post-processing manner. I am not saying that adversarial robustness is not important, but if that is the case, then the title and the main motivation of the paper should be revised. As for the evaluation of different input lengths, while it is an interesting perspective, I think we have multiple possible solutions for this problem in practice. For example, a detector trained with a text length at average length N can be used to detect long text using sampling or chunking. The latter can also alleviate the problem of detectors' poor performance in mixed content since now we are able to obtain scores at different parts of the input text. As such, the real value of this paper in providing a benchmark suit for real-world detector evaluation is somehow questionable.

There are also some soundness issues. For example, I wonder whether the back-translation and the human-written texts mixed with LLM-generated texts still conform to the original problem definition. The back-translation done by Google Translate might change the original meaning of the text, and the authors should at least introduce some kind of checking techniques. For the mixed scenario, the mixing proportion is a very important parameter. If it is aimed at mimicking AI-assisted academic writing, then it is perhaps beyond the scope of LLM-generated content detection. For example, AI-assisted writing is legal in some situations, but purely AI-generated content is often not.

Finally, the writing of the paper can be improved. This includes but is not limited to the following:
1. For section 3, I suggest improving Table 6 and splitting the detectors into zero-shot and supersized ones, just as Table 2 does for better visualization.

2. The authors should also provide more information on those supervised techniques (e.g., training data statistics) at the start of section 3.

3. Tables 2 and 3 should report the best performance instead of the average since some poor-performance methods could severely influence each category.

4. You seem to mix analyzing robustness issues in both sections 3.1 and 3.2. It might be better to move related contents from 3.1 to 3.2

**Strengths:**

+ A timely topic in the direction.
+ A new evaluation set that can contribute to a more systematic evaluation for LLM-generated content.

**Additional Feedback:**

1. What do you mean by white-box zero-shot detection at line 163?
2. What is the definition of the semantic attack mentioned at line 224?

**Clarity:**

More details should be provided for evaluated methods, experiment settings and important hyper-parameters.

**Documentation:**

Possibly yes.

**Ethics:**

No ethical concerns.

**Limitations:**

The major limitation is perhaps the novelty of the paper. Studying the influence of adversarial robustness and input length is not that so novel.

**Opportunities For Improvement:**

- The motivation (the main claim) should be revised, or the major experiment parts should be redesigned.
- The writing can be improved.
- Some conclusions need to be further justified.

**Relation To Prior Work:**

Yes

**Summary And Contributions:**

This paper introduces DetectEval, a benchmark that aims to explore the limitations of current state-of-the-art LLM-generated text detection methods. DetectEval involves data points collected from multi-domain and contains text written by both human and multiple generative LLMs. Additionally, this study further systematically benchmarks the robustness of both zero-shot and supervised detectors. Experiment results demonstrate that the current detectors are not yet capable of being used in complex real-world scenarios.

---

> ### Author Rebuttal · Authors · 2024-08-18
>
> We sincerely thank you for your insightful reviews of our manuscript.
>
> Below, we provide point-by-point responses to your comments and questions.
>
> ---
>
> **Correctness 3**: The attacks introduced in this study is not "sophisticated" to me.
>
> **Response**: Thank you for pointing out this important issue. We recognize that the use of the word "sophisticated" may indeed be inappropriate and could be misleading. We will replace it with **"meticulously designed"** to more accurately convey our approach, referring to our careful selection and crafting of attack scenarios that align with real-world scenarios.
>
> ---
>
> **Review Question 1 & Limitations & Correctness 1**: Major concern of the main claim of this paper, which states that DetectEval is a benchmark designed for real-world scenarios. The novelty of studying the influence of adversarial robustness and input length and the claim of real-world scenario.
>
> **Response**: We appreciate the reviewer's comments and would like to address your concerns by clarifying our contributions.
>
> Firstly, our adversarial methods were not fully explored by previous research. Moreover, this perspective is just one component of our framework. Our data construction encompasses a range of real-world use cases, including adversarial approaches, prompt attacks (detecting text generated by different instructions, as you mentioned), and data mixing scenarios. We selected our adversarial data methods based on real-world behaviors, such as non-native English speakers using translation tools, LLM-based text refinement, human-like word substitutions, and character-level spelling errors.
>
> Crucially, most of the work only discusses Task 1 of DetectEval rather than Task 2. This aspect is often overlooked in existing detection studies, especially for zero-shot methods, yet it's vital for real-world applicability. We evaluated the Adaptability Ability of detectors and call on researchers to focus on this evaluation perspective.
>
> While a small number of previous work has considered training and evaluation text lengths, our benchmark formally incorporates these factors. This ensures a comprehensive evaluation perspective and offers systematic analysis with interesting and novel insights.
>
> Furthermore, considering that detectors' misjudgment of human text may affect their credibility, we pioneered the inclusion of real-world human text impacts in our benchmark evaluation. By simulating human self-revision, writing errors, and data mixing, we analyze the potential impact of complex human text in the real world on detectors.
>
> ---
>
> **Review Question 1**: The issue of the evaluation of different input lengths.
>
> **Response**: We appreciate the reviewer's valuable insights and would like to further clarify potential misunderstandings. We strongly agree that the reviewer's suggestion of chunk-based detection is an excellent engineering approach for practical applications. However, this method requires determining the effective detection length for different detectors to define appropriate chunks. Moreover, for fine-grained detection, detectors need to perform well on shorter texts. Our Test-Time Evaluation Task covers text lengths ranging from 20 to 360 words (See Section 3.4, Figure 6), effectively assessing detector performance across various length intervals, especially for short texts (almost all detectors perform better on longer texts). Detectors that achieve higher scores in our Test-Time Evaluation Task will undoubtedly demonstrate better and more stable performance in chunk-based detection approaches and various application scenarios. As such, evaluating detector robustness across different text lengths is crucial for real-world application scenarios.
>
> ---
>
> **Review Question 2**: The back-translation done by Google Translate might change the original meaning of the text, and the authors should at least introduce some kind of checking techniques.
>
> **Response**: Thank you for your suggestion. We will use BERTScore [8], an automatic evaluation metric that assess the translation quality from the perspective of semantics. The specific results are as following table, which show that the texts before and after back-translation maintain good semantic consistency, with BERTScore (Avg. F1-scores) **all exceeding 0.90 among 4 domains**, validating the reasonableness of our use of back-translation done by Google Translate.
>
> The BERTScore checking techniques will be updated in the framework to ensure semantic consistency of the data before and after back-translation.
>
> | Domains          | BERTScore (Average F1) |
> |-------------------------|-----------------|
> | Academic Writing | 0.9243         |
> | News Writing       | 0.9233          |
> | Creative Writing | 0.9195            |
> | Social Media     | 0.9268             |
>
> **Review Question 3**: The issue of the mixing proportion and mimicking AI-assisted academic writing.
>
> **Response**: In our data construction framework, we will include the mixing proportion as a variable to allow researchers to independently create more challenging scenarios.
>
> We chose the 0.25 ratio to strike a balance between challenge difficulty and analyzability. It's important to note that this scenario isn't meant to simulate AI-assisted academic writing. Rather, it challenges the real-world situation where people directly incorporate AI-generated text into their work with minimal editing. This could involve adding a small portion (<25%) of content from existing sources or their own input to AI-generated text (similar to making minor edits to an AI-generated paper). In such cases, the resulting text is likely to be of low quality and lack credibility, which should be identifiable and subject to appropriate scrutiny.

---

> ### Author Rebuttal · Authors · 2024-08-18
>
> **Review Question 4**: The writing of the paper can be improved.
>
> **Response**: Thank you for the constructive comments. We have revised the relevant writing sections of the paper according to the reviewer's suggestions as follows:
>
> 1. Thanks for the suggestion, but considering that Figure 6 only shows a supervised detector, it may be difficult to balance the visualization of the table by splitting like Figure 2. To enhance readability, we've removed NPR from the table to maintain an appropriate font size and added labels indicating detector types next to the corresponding methods. (see **Table 2 in the attached PDF**)
> 2. Actually, the training data volume and parameter settings for Supervised techniques have already been described in **Page4, Table 1 Benchmark Statistics** and **Appendix F.2 Detectors Settings**. We'll add this crucial information into Section 3 for improved clarity.
> 3. We will report both best and average scores. The best score will reflect the upper limit of detection capabilities for each category, while the average score will show how most detectors perform in that category. We believe this approach will provide readers with a more intuitive understanding of the current challenges and trends in the field.
> 4. We will move the robustness-related discussion from section 3.1 to 3.2, which can make the logic of our manuscript clearer.
>
> We will add the above revisions in the final version and thanks for your valuable suggestions.
>
> ---
>
> **Correctness 2**: The issue of conclusion that training detectors on shorter text is more effective.
>
> **Response**: Thank you for the insightful comments. To help validate this point, we conducted an evaluation using the latest llama-3-8B model (as our experimental equipment struggles to support inference for 13B and larger models). The results can be seen in **Table 4 of the attached PDF**. It is evident that using the Llama-3-8B as a substitute model maintains a consistent trend with the results of gpt-neo-2.7B., strongly validating our manuscript's assertion that "Training detectors on shorter text is more effective." Additionally, regarding zero-shot methods, according to the findings of Mireshghallah et al. [9], larger detectors do not necessarily learn better, which to some extent supports the results obtained using gpt-neo-2.7B.
>
> ---
>
> **Correctness 4**: The authors should further justify that how DetectEval provide a data curation framework.
>
> **Response**: We will include detailed instructions for using DetectEval in practice in the Appendix. Currently, we have made our benchmark data and code framework publicly available in an anonymous repository (https://anonymous.4open.science/r/DetectEval/). Our data generation pipeline is designed to be highly integrated, enabling the swift creation of new challenging datasets. Users can easily customize the process by specifying the data path for enhancement and selecting the desired attack type. This streamlined approach is implemented through our data_generation.sh and benchmark_construction.sh scripts, which require only minimal parameter adjustments.
>
> We will update the framework with more detailed usage instructions and include this in the final version.
>
> ---
>
> **Clarity**: More details should be provided for evaluated methods, experiment settings and important hyper-parameters.
>
> **Response**: Thank you for your great suggestion. We will move the details of evaluated methods, experiment settings, and important hyper-parameters from **section F.2 Detectors Settings** to the main text to improve the clarity of the article. We will also add detailed guidelines to encourage other detector researchers to submit evaluation their results, to keep the leaderboard up-to-date and cutting edge. We will include this part in the final version.
>
> ---
>
> **Additional Feedback**: What do you mean by white-box zero-shot detection and the definition of the semantic attack?
>
> **Response**:
>
> - For **white-box zero-shot detection**: To clarify, most current zero-shot detectors rely on accessing the logits and their variants from a surrogate model to construct decision thresholds for identifying LLM-generated text. According to reviews by Yang et al. [10], these approaches are classified as white-box zero-shot detection. However, recent developments in zero-shot detection, such as RAIDAR [11] and Revise [12], do not depend on surrogate model logits or their variants. These newer methods have been categorized as black-box zero-shot detection.
> - For **Semantic Attack**: Thank you for pointing out this important error. This is indeed a writing error, and we will correct "semantic attack" to "paraphrase attack". This revision will be include in the final version.
>
> ---
>
> # References
> Please see **Rebuttal References** in the global response.

---

> ### Author Response · Authors · 2024-08-27
> **Reply to Reviewer tJNz**
>
> Dear Reviewer tJNz,
>
> Thanks again for the insightful review. We have provided detailed clarifications on your questions point to point. As the discussion period is nearing its end , we would like to see if there is any additional information you require or if there are any concerns we can address. We sincerely hope you can appreciate the significance of our work and consider updating your rating.
>
> Thank you for your time and consideration!
>
> Best wishes,
>
> The Authors

---

> > ### Comment · Reviewer_tJNz · 2024-08-27
> >
> > Thank you for the additional details regarding your experiments, especially the new results based on BERTScore. As the rebuttal has addressed some of my concerns, I will raise my rating to 6 accordingly.

---

> > > ### Author Response · Authors · 2024-08-27
> > > **Response to Reviewer tJNz**
> > >
> > > Dear Reviewer tJNz,
> > >
> > > We are delighted that our response addresses some of your concerns. Again, we thank you for your valuable feedback and comments.
> > >
> > > Please let me know if you have any further questions or concerns for us, we would be very happy to discuss them with you.
> > >
> > > Best,
> > >
> > > Authors

---

### Official Review · Reviewer_rVaJ · 2024-07-24

**Rating:** 5
**Confidence:** 4

**Review:**

Pros:
1. The paper considers an important problem of constructing a unified evaluation framework for assessing LLM text detectors.
2. The constructed dataset is sufficiently big and covers a number of domains and language models.
3. The tasks considered in the paper are moderately challenging including detecting LLM generated text processing with various attacks.

Cons:
1. While paper benchmarks existing LLM text detectors against their benchmark, the authors do not include recent state-of-the-art detectors, such as Binoculars [1] and Ghostbuster [2].
2. The presentation of the benchmark results could be improved. Most of the tables (Tables 1-4) are not readable because of the font size, and the same applies to Figures 2-5.  Also, the paper does not provide details on how exactly LLM generated texts are constructed for the dataset.

[1] Hans, A., Schwarzschild, A., Cherepanova, V., Kazemi, H., Saha, A., Goldblum, M., Geiping, J. and Goldstein, T., 2024. Spotting llms with binoculars: Zero-shot detection of machine-generated text. ICML 2024.
[2] Verma, V., Fleisig, E., Tomlin, N. and Klein, D., 2023. Ghostbuster: Detecting text ghostwritten by large language models. arXiv preprint arXiv:2305.15047.

**Strengths:**

Please, see above.

**Additional Feedback:**

Please see my comments and questions above.

**Clarity:**

The paper is mostly well written, however details on dataset generations could be stated more clearly.

**Correctness:**

Evaluating the robustness of LLM text detectors is an important aspect, however, it is unclear why perturbed texts, such as paraphrased texts or texts with inserted human-written sections, should be classified as LLM-generated. Could the authors please clarify this point? My understanding is that LLM text detectors are intended to identify "purely generated texts" rather than texts where language models are used for assistance for example.

**Documentation:**

Yes

**Limitations:**

Yes

**Opportunities For Improvement:**

1. Including more state-of-the-art approaches for LLM text detection in benchmark would improve the paper.
2. Improving presentation of the results, and clarifying how exactly data was generated for the benchmark would make it easier to read the paper.

**Relation To Prior Work:**

Yes

**Summary And Contributions:**

The paper proposes a benchmark for evaluating LLM generated text detectors. The authors collected a dataset of machine generated and human written texts from multiple domains (academic writing, news writing, creative writing, and social media) and multiple LLMs. The authors also created challenging scenarios with different attacks making it harder to detect llm generated texts. Finally, the authors evaluated a number of existing LLM text detectors and identified a gap in their performance.

---

> ### Author Rebuttal · Authors · 2024-08-18
>
> We thank the reviewer for the time and expertise you have invested in these reviews.
>
> Below, we provide point-by-point responses to your comments and questions.
>
> ---
>
> **Opportunities For Improvement 1**: Including more state-of-the-art approaches for LLM text detection in benchmark would improve the paper.
>
> **Response**: This is a great suggestion. Based on your recommendation, we have added the experimental results for Binoculars [6], Revise [12] and DNA-GPT [13]. We are delighted to share that Binoculars achieved the latest SOTA performance among all zero-shot methods in our benchmark, outperforming the second-best by 15% on average and maintaining top performance across all evaluation tasks (all over 70%). Revise secured the second-best performance, demonstrating greater robustness to text length variations compared to Log-Rank, which ranked third, although it performed less well on Real-World Human Writing Scenarios. (See the **updated leaderboard in Table 3 in the attached PDF**.)
>
> Furthermore, we are in the process of evaluating other recent SOTA detectors. Due to the substantial time and resources needed (as each detector requires approximately 34 sets of experiments to assess thoroughly and some detectors are quite resource-intensive), we haven't yet completed all the tests. So far, **we have updated the results for 3 new recent SOTA detectors and will continue to provide you with the latest experimental findings**.
>
> We also plan to create a leaderboard and allow individuals or institutions to submit detection results for their newest methods. We will include this part in the final version.
>
> ---
>
> **Opportunities For Improvement 2**: The presentation of the benchmark results could be improved. Most of the tables (Tables 1-4) are not readable because of the font size, and the same applies to Figures 2-5.
>
> **Response**: We greatly appreciate your reviewer's insightful suggestion to enhance the article's readability. In response, we have adjusted the font sizes and distribution, and deleted unnecessary spaces of Tables 1-4 and Figures 2-5, resulting in clearer and more concise visual presentations. We've included a revised version of Table 4 (see **Table 2 in the attached PDF**) as an example of these enhancements.
>
> We will include these enhancements in the final version.
>
> ---
>
> **Clarity**: The paper does not provide details on how exactly LLM generated texts are constructed for the dataset.
>
> **Response**: Actually, we have already provided details of data generation (As we described in **Appendix E Data Collection** of our manuscript), but due to space limitations in the main article, it was placed in the appendix of our submitted paper.
>
> We will consider these details and reflecting corresponding content in the main body of the paper.
>
> ---
>
> **Correctness**: Why perturbed texts, such as paraphrased texts or texts with inserted human-written sections, should be classified as LLM-generated.
>
> **Response**: We clarify this point here. While current LLM text detectors primarily focus on identifying "pure LLM-generated texts," it's important to note that such texts are rarely used in their raw original form in real-world scenarios. This is often due to efforts to enhance content quality (make them more human-like) or evade detection. Consider this: simply performing character-level perturbations on a "pure LLM-generated texts," paraphrasing certain segments, or manually adding small portions of content can evade detector identification.
>
> In our paper, simple perturbation attacks hurt the detectors’ performance, as shown in Table 3 and Table 13. Therefore, the construction of DetectEval aims to encourage detectors to improve their ability to handle such real-world scenarios, which has also received attention in many recent works [7].
>
> On the other hand, we strongly support the responsible use of language models as writing assistants. When humans use LLMs to refine or polish their own writing, it can be highly beneficial. However, it's crucial to distinguish this from the misuse of LLMs or their slightly modified versions. Such practices, like when students use LLMs to draft entire papers and then make only minor edits, are often counterproductive. Instead, we should encourage practices where students write their own drafts and then use LLMs for refinement.
>
> Consequently, it's important that detectors can accurately identify perturbed or paraphrased texts derived from purely generated content. In the age of LLMs, these detectors serve a vital role in assessing the credibility of content. This capability is not about discouraging LLM use altogether, but rather about promoting responsible usage and maintaining academic and professional integrity. This is also an important initial motivation for us to develop DetectEval.
>
> ---
>
> # References
> Please see **Rebuttal References** in the global response.

---

> ### Author Response · Authors · 2024-08-27
> **Reply to Reviewer rVaJ**
>
> Dear Reviewer rVaJ,
>
> Thanks again for the insightful review. We have provided detailed clarifications on your questions point to point. As the discussion period is nearing its end , we would like to see if there is any additional information you require or if there are any concerns we can address. We sincerely hope you can appreciate the significance of our work and consider updating your rating.
>
> Thank you for your time and consideration!
>
> Best wishes,
>
> The Authors

---

> ### Author Response · Authors · 2024-08-30
> **Response to Reviewer rVaJ**
>
> Dear Reviewer rVaJ,
>
> We sincerely appreciate the reviewer’s constructive suggestions and believe that the additional experiments and explanations significantly improve the quality of our submission. As the discussion period is nearing its end, we would like to see if there is any additional information you require or if there are any concerns we can address.
>
> Thank you for your time and consideration!
>
> Bests,
>
> The Authors

---

### Official Review · Reviewer_7jmF · 2024-08-03
**Benchmarking LLM-Generated Text Detection**

**Rating:** 8
**Confidence:** 3
**Clarity:** The paper is well written and an Appe…

**Review:**

This work provides a novel benchmark to evaluate LLM-generated-text detectors. The performance of several SOTA detectors is assessed and various analysis is done to examine the contributed factors.

I appreciate the effort in curating a real-world like dataset and the extensive analysis done under different criteria. The publicly available framework will be a definite benefit for future researches.

**Strengths:**

* The study provides the  data curation framework and open-sourced
* The benchmark is designed to resemble the real-world applications closely
* Several common LLMs are considered and an elaborative analysis is given

**Additional Feedback:**

I enjoyed the paper. The analysis is done extensively. Some additional information as I mentioned under the Improvements and Limitations would make the work outstanding, according my opinion.

**Correctness:**

The dataset construction is sound. Various attacks (Prompt, Paraphrase, etc.) are used to generate the data, aiming to simulate complex real-world detection scenarios.

**Documentation:**

The datasets used are completely open source and public to the research community.

**Ethics:**

No violation is found

**Limitations:**

The study evaluates both Zero-shot and supervised detectors under specific factors. Despite the results of the evaluation criteria, it sounds like the superiority of Supervised over Zero-shot may be challenged in the future due to the rapid advancement of new LLMs. If it can be challenged, it should be explicitly mentioned (not vaguely) under the limitations.

**Opportunities For Improvement:**

No evidence was given to show how the proposed benchmark differs from the existing benchmarks

**Relation To Prior Work:**

Popular benchmarks are referred to. Also, the references for Data sources, Models, and Data generation were cited.

**Summary And Contributions:**

DetectEval is a novel benchmark designed to evaluate the effectiveness and performance of both Zero-shot and Supervised detectors against LLM-generated text. The study analyzes the potential impact of writing styles, model types, attack methods, text lengths, and attacked human-written texts on different types of detectors. Various matrices are presented that showcase the superior performance of Supervised detectors over Zero-shot.

---

> ### Author Rebuttal · Authors · 2024-08-18
>
> We thank the reviewer for the time and expertise you have invested in these reviews. We are delighted to receive positive feedback that our work provides a solid contribution to the field.
>
> Below we provide point-by-point responses to your comments and questions.
>
> ------
>
> **Opportunities For Improvement**: No evidence was given to show how the proposed benchmark differs from the existing benchmarks
>
> **Response**: Thank you for your valuable input. Actually, We've described the differences between our work and existing studies in the second paragraph of Section 1 (lines 26-29).
> Specifically, prior studies such as TuringBench [1], MGTBench [2], MULTITuDE [3], MAGE [4] and M4 [5] have predominantly focused on evaluating detector performance across various domains, generative models, and languages using idealized test data. However, these investigations have not sufficiently explored the assessment of detectors' capabilities in real-world application scenarios. In contrast, our work addresses this gap by evaluating detector performance through simulations of common scenarios encountered in practical applications.
>
> To further clarify our article, we have compiled a **comparative table** highlighting the differences between DetectEval and existing benchmarks. (See **Table 1 in the attached PDF**.)
>
> ------
> **Limitations**: If Supervised detectors over Zero-shot may be challenged in the future due to the rapid advancement of new LLMs, it should be explicitly mentioned (not vaguely) under the limitations.
>
> **Response**: Thank you for your valuable feedback. While it's true that Supervised detectors experience performance drops when confronted with new domains or models, In our experiments, we've observed similar limitations in zero-shot detectors.
> Although zero-shot detectors can potentially benefit from diverse data styles and generator model types, our DetectEval results (please refer to Section 3.2, Table 4; Appendix, Table 6) demonstrate their significant unreliability compared to Supervised detectors. Given the current state of Zero-shot methods, Supervised detectors are more likely to remain effective when facing new models.
>
> We will expand the limitations section to **clearly point out** that the effectiveness and robustness of existing Supervised detectors compared to Zero-shot methods in real-world scenarios.
>
> ---
>
> # References
> Please see **Rebuttal References** in the global response.

---

### Author Rebuttal · Authors · 2024-08-18

# Dear Reviewers,

We appreciate all the reviewers for their constructive and insightful feedback! We sincerely thank you for taking the time and effort to review our work. We have provided detailed responses to each reviewer's comments and questions in the various author replies. All revisions will be reflected in the final version.

Below are the references and attached tables and figures mentioned in our responses.

---

# Rebuttal References

[1] Uchendu, A., Ma, Z., Le, T., Zhang, R., & Lee, D. (2021, November). TURINGBENCH: A Benchmark Environment for Turing Test in the Age of Neural Text Generation. In Findings of the Association for Computational Linguistics: EMNLP 2021 (pp. 2001-2016).

[2] He, X., Shen, X., Chen, Z., Backes, M., & Zhang, Y. (2023). Mgtbench: Benchmarking machine-generated text detection. arXiv preprint arXiv:2303.14822.

[3] Macko, D., Moro, R., Uchendu, A., Lucas, J., Yamashita, M., Pikuliak, M., ... & Bieliková, M. (2023, December). MULTITuDE: Large-Scale Multilingual Machine-Generated Text Detection Benchmark. In Proceedings of the 2023 Conference on Empirical Methods in Natural Language Processing (pp. 9960-9987).

[4] Li, Y., Li, Q., Cui, L., Bi, W., Wang, Z., Wang, L., ... & Zhang, Y. (2024, August). Mage: Machine-generated text detection in the wild. In Proceedings of the 62nd Annual Meeting of the Association for Computational Linguistics (Volume 1: Long Papers) (pp. 36-53).

[5] Wang, Y., Mansurov, J., Ivanov, P., Su, J., Shelmanov, A., Tsvigun, A., ... & Nakov, P. (2024, March). M4: Multi-generator, Multi-domain, and Multi-lingual Black-Box Machine-Generated Text Detection. In Proceedings of the 18th Conference of the European Chapter of the Association for Computational Linguistics (Volume 1: Long Papers) (pp. 1369-1407).

[6] Hans, A., Schwarzschild, A., Cherepanova, V., Kazemi, H., Saha, A., Goldblum, M., ... & Goldstein, T. Spotting LLMs With Binoculars: Zero-Shot Detection of Machine-Generated Text. In Forty-first International Conference on Machine Learning.

[7] Krishna, K., Song, Y., Karpinska, M., Wieting, J., & Iyyer, M. (2024). Paraphrasing evades detectors of ai-generated text, but retrieval is an effective defense. Advances in Neural Information Processing Systems, 36.

[8] Zhang, T., Kishore, V., Wu, F., Weinberger, K. Q., & Artzi, Y. BERTScore: Evaluating Text Generation with BERT. In International Conference on Learning Representations.

[9] Mireshghallah, N., Mattern, J., Gao, S., Shokri, R., & Berg-Kirkpatrick, T. (2024, March). Smaller Language Models are Better Zero-shot Machine-Generated Text Detectors. In Proceedings of the 18th Conference of the European Chapter of the Association for Computational Linguistics (Volume 2: Short Papers) (pp. 278-293).

[10] Yang, X., Pan, L., Zhao, X., Chen, H., Petzold, L., Wang, W. Y., & Cheng, W. (2023). A survey on detection of llms-generated content. arXiv preprint arXiv:2310.15654.

[11] Mao, C., Vondrick, C., Wang, H., & Yang, J. Raidar: geneRative AI Detection viA Rewriting. In The Twelfth International Conference on Learning Representations.

[12] Zhu, B., Yuan, L., Cui, G., Chen, Y., Fu, C., He, B., ... & Gu, M. (2023, December). Beat LLMs at Their Own Game: Zero-Shot LLM-Generated Text Detection via Querying ChatGPT. In Proceedings of the 2023 Conference on Empirical Methods in Natural Language Processing (pp. 7470-7483).

[13] Yang, X., Cheng, W., Wu, Y., Petzold, L. R., Wang, W. Y., & Chen, H. DNA-GPT: Divergent N-Gram Analysis for Training-Free Detection of GPT-Generated Text. In The Twelfth International Conference on Learning Representations.

---

# Rebuttal Tables and Figures

Please download the attached PDF file.

---

### Decision · Program_Chairs · 2024-09-26

**Decision:**

Accept (Poster)

**Comment:**

This paper describes DetectEval, a benchmark for evaluating LLM detectors. The authors curated a significant number of text samples from a variety of contexts, using a number of popular LLMs, and used them as a benchmark to test LLM detectors.  As a whole, this is a much needed benchmark in a fast moving area of extremely high importance (to educators trying to detect plagiarism, publishers seeking to weed out AI-generated manuscripts, and reviewers trying to identify AI-generated literature).  This benchmark covers multiple realistic scenarios and relies on popular LLM models, making its results highly relevant.

Pros
- clearly written paper on an extremely important and high impact topic
- the authors were extremely responsive to reviewers in addressing weaknesses and adding significant results to the paper, e.g. adding Binoculars to the detectors evaluated
- the focus on adversarial robustness is very much appreciated. Given the incentives in LLM-related scenarios, it is quite feasible to consider cases where someone will implement easy to use counterattacks on detectors, and make them available (for a fee) to LLM users seeking to avoid detection.

Cons
- Some of the figures and tables are simply too small to read, even in digital form. Authors have to make some decisions about which results to include and which to put into an extended version for arxiv

In summary, the paper makes a solid contribution to a very important space. The benchmark should provide real utility to researchers on the topic and practitioners looking for consistent evaluation methods.

One final note, I would suggest changing the system name, because nothing in DetectEval indicates it's a benchmark for LLMs. Given the # of benchmarks for LLMs, diffusion models, and all sort of AI systems, DetectEval is bound to cause confusion. It'd be better to include some mention of LLMs in its name to distinguish it from unrelated benchmarks.